# From Waste to Worth: Upcycling Plastic into High-Value Carbon-Based Nanomaterials

**DOI:** 10.3390/polym17010063

**Published:** 2024-12-30

**Authors:** Ahmed M. Abdelfatah, Mohamed Hosny, Ahmed S. Elbay, Nourhan El-Maghrabi, Manal Fawzy

**Affiliations:** 1Green Technology Group, Faculty of Science, Alexandria University, Alexandria 21511, Egypt; ahmedmohamedfatah@alexu.edu.eg (A.M.A.); ahmed.elbay@alexu.edu.eg (A.S.E.); nourhan.elmaghrabi@alexu.edu.eg (N.E.-M.); manal.fawzy@alexu.edu.eg (M.F.); 2National Biotechnology Network of Expertise (NBNE), Academy of Scientific Research and Technology (ASRT), Cairo 11694, Egypt

**Keywords:** plastic waste, carbon-based nanomaterials, graphene, waste valorization, environmental sustainability, upcycling

## Abstract

Plastic waste (PW) presents a significant environmental challenge due to its persistent accumulation and harmful effects on ecosystems. According to the United Nations Environment Program (UNEP), global plastic production in 2024 is estimated to reach approximately 500 million tons. Without effective intervention, most of this plastic is expected to become waste, potentially resulting in billions of tons of accumulated PW by 2060. This study explores innovative approaches to convert PW into high-value carbon nanomaterials (CNMs) such as graphene, carbon nanotubes (CNTs), and other advanced carbon structures. Various methods including pyrolysis, arc discharge, catalytic degradation, and laser ablation have been investigated in transforming PW into CNMs. However, four primary methodologies are discussed herein: thermal decomposition, chemical vapor deposition (CVD), flash joule heating (FJH), and stepwise conversion. The scalability of the pathways discussed for industrial applications varies significantly. Thermal decomposition, particularly pyrolysis, is highly scalable due to its straightforward setup and cost-effective operation, making it suitable for large-scale waste processing plants. It also produces fuel byproducts that can be used as an alternative energy source, promoting the concept of energy recovery and circular economy. CVD, while producing high-quality carbon materials, is less scalable due to the high cost and required complex equipment, catalyst, high temperature, and pressure, which limits its use to specialized applications. FJH offers rapid synthesis of high-quality graphene using an economically viable technique that can also generate valuable products such as green hydrogen, carbon oligomers, and light hydrocarbons. However, it still requires optimization for industrial throughput. Stepwise conversion, involving multiple stages, can be challenging to scale due to higher operational complexity and cost, but it offers precise control over material properties for niche applications. This research demonstrates the growing potential of upcycling PW into valuable materials that align with global sustainability goals including industry, innovation, and infrastructure (Goal 9), sustainable cities and communities (Goal 11), and responsible consumption and production (Goal 12). The findings underscore the need for enhanced recycling infrastructure and policy frameworks to support the shift toward a circular economy and mitigate the global plastic crisis.

## 1. Introduction

The global production of plastics has seen a steep rise over the years. In 2019, nearly 370 million tons of plastic were produced worldwide including thermoplastics, thermosets, polyurethanes, adhesives, coatings and sealants, elastomers, and PP-fibers. (polyacrylic fibers, PET fibers, and PA fibers are not included) with around 15.7% coming from Europe [1]. By 2023, the amount of plastic generated surged to 159 million tons, with 43% reportedly contributing to land pollution [2].

In another study by 2021, the commercial production of plastic increased up to 390.7 million tons globally [3]. Plastic products, though indispensable in modern life, contribute significantly to irreversible environmental harm. Designed with a durability of up to 50 years, plastics were intended for long-term use. However, the rise of a “throw-away” culture has driven a rapid increase in single-use plastic production, outpacing all other sectors. Each year, approximately 284 million tons of PW are generated, with over 2.41 million tons ending up in the ocean. Through chemical and physical processes, these plastics break down into microplastic particles smaller than 5 mm, releasing over 10 million tons into the environment annually [4,5]. Of this waste, 36.4% is expected to end up in landfills or the environment, another 36.4% is likely to be incinerated, and only 27.2% is anticipated to be recycled [4]. The EU Directive 94/62/EC requires EU countries to recycle 50% of plastic by 2025 and 55% by 2030. However, countries cannot achieve this target; for example, Germany, a country that has a modern and well-established PW recycling system in the form of the dual system, only achieves recycling rates of less than 30% for post-consumer plastics. This demonstrates the limitations of mechanical recycling in meeting the ambitious EU goal for plastic recycling. Therefore, new recycling concepts and technologies are desperately needed to complement or substitute traditional recycling to increase the circularity of PW and achieve recycling targets [6].

Globalization has fragmented supply chains, complicating the assessment of life-cycle environmental impacts, including waste management. Since 2019, approximately five million tons of PW have been traded annually, predominantly from high-income to low-income countries. This practice raises concerns about mismanagement, especially in the Global South, where inadequate waste treatment systems contribute to river pollution and the “plastic soup” in oceans [7]. For instance, over half of Indonesia’s PW is incinerated without energy recovery, and more than 60% of marine plastic emissions come from countries like the Philippines, India, Malaysia, and Indonesia [8].

In 2018, China’s ban on PW imports redirected exports to other countries such as Malaysia, Indonesia, Turkey, and Vietnam [9]. This shift further stressed the waste management systems of these countries. In response, the EU has proposed banning exports to non-OECD countries, emphasizing domestic responsibility for proper waste treatment [10].

Although global recycling rates for plastics remain below 10% [4], there is an assumption that traded plastic is primarily recycled. However, determining recycling rates in receiving countries is challenging due to data limitations. Studies often rely on assumed recycling rates, ranging from 10% to 90%, highlighting the need for robust, transparent data. For example, environmental impacts were estimated using assumed recycling rates of 10–40% for Southeast Asia [9], while another study [8] used rates ranging from 8.7–50% for the USA. Similarly, a recent study [11] relied on broad recycling rate estimates of 50–90% for exported plastics from Europe.

The recycling rate of imported PW cannot simply be equated with domestic averages due to key differences: imported plastics are often pre-selected and uniform, while domestic waste is more heterogeneous and challenging to recycle. Additionally, the UN Comtrade database indicates that importing countries pay for PW, demonstrating its economic value (Figure 1) [12]. If imported waste were primarily dumped or burned, it would lead to significant financial losses for importers, making such practices unsustainable. This suggests that at least part of the imported plastic is processed into valuable recyclates to offset initial costs. Comprehensive data and policies are essential to ensure sustainable and effective global PW management.

These predictions demonstrate the ongoing issue of mismanagement, largely due to the limitations of current recycling and reuse technologies as they can only process certain types of PW, which leaves much of this waste discarded in landfills. Moreover, traditional recycling can only be done for a limited time otherwise the produced plastic loses its quality. Recycled plastic often has lower quality than new ones and it requires pretreatment steps of cleaning, careful sorting, and drying, which is time-consuming and requires additional cost leading to inadequate and ineffective recycling rates [13,14]. Although plastics have greatly benefited human society, they also cause significant environmental harm. Designed to last for up to 50 years, many plastics are now part of the growing single-use culture. Each year, 284 million tons of PW are generated, with over 2.41 million tons ending up in the ocean. The majority of oceanic plastic is from land sources as between 70% and 80% of its weight is carried by rivers or coastlines from land to the sea [14]. The remaining 20% to 30% originates from marine sources, including lines, ropes, abandoned ships, and fishing nets [15]. Through chemical and physical processes, this waste breaks down into microplastics smaller than 5 mm, releasing over 10 million tons of these pollutants into the environment annually [5,16]. Microplastics can severely affect human health as they can enter the body through ingestion, skin contact, and inhalation [17]. The potential harmful impacts come from the leaching of its inorganic and organic toxic chemical components and its ability to adsorb hazardous extraneous substances through the biological or chemical vectors associated with it. Moreover, it can cause direct physical damage from ingested particles of plastic debris that attach to various organs [18]. As a result, PW is increasingly polluting soil, freshwater, and oceans globally, causing significant harm to living organisms and natural ecosystems. To address the global PW crisis, both preventive strategies aimed at reducing the release of plastic into the environment (upstream measures) and mitigation efforts to lessen its impact once it has been released (downstream measures) have been explored. Effective waste management is a key component of upstream strategies. However, recycling efforts face numerous challenges, including the high cost of recycling PW and the heterogeneity of PW that necessitates a preprocessing step to sort out different types of PW [19,20].

Upcycling reduces waste accumulation while simultaneously creating economic opportunities by transforming discarded materials into higher-value products, making it an effective solution to address the challenges posed by PW. In this context, utilizing PW as a raw material for producing value-added products and materials has been proposed to enhance the economic viability of plastics recycling. Recent studies have moved beyond the traditional use of PW in bitumen [21], and food production [22] to more advanced applications. Such applications include the generation of gaseous and liquid fuels, as well as valuable chemicals [23,24]. Upcycling plastic waste (PW) into carbon-based materials with diverse nanostructures and morphologies such as hierarchical porous carbon, carbon nanotubes (CNTs), graphene, carbon quantum dots, and carbon-based composites [25,26,27,28,29,30,31,32,33,34] offers significant utility in various applications, including water treatment.

Polyethylene terephthalate (PET) can be upcycled to various bio-polymers, and in this regard, PET waste was converted to polyhydroxyalkanoate through thermal hydrolysis followed by degrading the produced terephthalate microbiologically using *Pseudomonas umsongensis* GO16 [35]. Also, they used the same bacterial strain to produce bio-based poly(amide urethane), which was a new bioplastic. Polystyrene (PS) is another common plastic material that could be upcycled to bio-oils owing to its high yield of that sort of oils that can be up to 84 wt% via pyrolysis [36]. Such a yield is substantially higher than that of other types of plastics such as polyethylene (PE) with about 43 wt% of bio-oil yield [37]. Polyvinyl chloride (PVC) could be dechlorinated to produce high-density polyethylene (HDPE) waxes through a heterogeneous catalytic reaction using a carbon-supported platinum catalyst [38]. These waxes are more valuable than PVC itself. Upcycling of polypropylene (PP) could be useful in water treatment as proved by [39] where pp was co-pyrolyzed with a dried biomass sample of *Microcystis aeruginosa* using potassium carbonate as an activator to produce a cyanobacteria-plastic carbon composite. This composite showed promising results in adsorbing methylene blue with an adsorption capacity of 490 mg/g owing to its high surface area (2135 m^2^/g). Co-pyrolysis has also been used to produce carbon materials such as biochar with a high rate of bio-oil production by pyrolyzing biomass such as pinecone with PW such as low-density polyethylene (LDPE), PP, and PS at 500 °C [40]. Through a three-step process comprising air-bromination, dehydrobromination, and olefin metathesis reactions, PE was upcycled to α,ω-divinyl-functionalized oligomers, which can be utilized in producing chemical additives, lubricants, waxes, and plastics as well [41].

Carbon-based nanomaterials are widely recognized as the most used electrode materials in electrochemical capacitors. This is attributed to their exceptional surface area, diverse structural configurations, excellent electrical conductivity, and highly porous textures. Similarly, nickel cobalt manganese (NCMs) stand out due to their remarkable thermal and chemical stability, tunable surface chemistry and morphology, and broad operating voltage range. Additionally, their nontoxic nature, cost-effectiveness, and widespread availability at the commercial scale further position NCMs as promising candidates for supercapacitor electrode materials [42].

The incorporation of waste plastic into Li-ion batteries significantly reduces their production costs, facilitating their broader application in electric buses, cars, and large-scale renewable energy systems such as wind and solar power plants. For instance, non-biodegradable polyethylene plastics (e.g., shopping bags), which have a short useful life, can be repurposed as anode materials in Li-ion batteries through sulfonation and pyrolysis. During the carbonization process, sulfonated polyethylene is cross-linked at approximately 500 °C and subsequently decomposed into carbon chips. These finely ground carbon chips, comprising about 80% carbon derived from polyethylene, are effectively utilized as anode materials, demonstrating a sustainable and efficient use of waste plastics [43].

Various materials, including carbon-based materials, metal oxides, and conducting polymers, are used in supercapacitors. Among these, carbon-based materials are the most effective choice for energy storage devices due to their superior performance. Activated carbon is the most utilized carbon-based material for capacitor electrodes, thanks to its large specific surface area (>1000 m^2^/g), high pore volume, affordability, and excellent electrical properties. It achieves high capacitance values in both organic (120 F/g) and aqueous (115–340 F/g) electrolytes, with some exceeding 200 F/g. The physicochemical properties of activated carbon, such as surface area, porous structure, and pore size distribution, are strongly influenced by the carbon precursors and activation methods employed. Enhancing the specific surface area has been identified as a promising strategy for improving capacitance. For example, a significant increase in capacitance from 17.68 F/g to 171.2 F/g by increasing the specific surface area from 621 m^2^/g to 2685 m^2^/g was recently reported [44,45]. Activated carbon, produced through the physical or chemical activation of solid waste (carbon precursors), typically exhibits narrow micropores (<0.5 nm) and limited porous pathways. These structural limitations impede electrolyte ion transport, particularly in organic electrolytes, leading to reduced capacitance. To address this limitation, carbon-based materials with mesopores (2–50 nm), known as well-ordered mesoporous carbons, are synthesized using soft or hard template approaches. These materials exhibit excellent electrochemical performance at high current densities due to improved ion accessibility, though their lower surface areas negatively impact capacitance. As an alternative, zeolite-templated carbon (ZTC), characterized by an intricate pore network and a large specific surface area (SSA) of approximately 3000 m^2^/g, can significantly enhance capacitance by leveraging its high SSA. Considering the complementary roles of pore sizes, mesopores, macropores facilitating ion diffusion at high current densities, and micropores enhancing capacitance nickel cobalt manganese (NCMs) with well-interconnected micro-mesoporous structures represent a highly promising option for electrochemical capacitors (ECs).

Carbon nanodots (CNDs) are typically described as dispersed, quasi-spherical nanoparticles with a multi-layered graphene inner core and numerous oxygen- or nitrogen-containing functional groups along their edges [46]. These smart carbon nanomaterials have garnered significant attention due to their versatile applications across various fields, particularly as fluorescent materials for biomarker detection, including those associated with magnetic nanoparticles (MNPs). CNDs exhibit exceptional properties such as high chemical stability, low toxicity, and excellent water solubility, making them environmentally safe alternatives to semiconductor quantum dots [47]. In recent decades, CNDs have been extensively studied for applications in biosensing, owing to their zero-dimensional structure, optical activity, and size below 10 nm [48]. There has also been a growing interest in developing advanced sensor platforms utilizing CNDs [49]. For example, co-reactant-doped nanomaterials were introduced to enhance electroluminescence (ECL) performance. Their work demonstrated the role of amine-functionalized carbon quantum dots (f-CQDs) as co-reactants, significantly improving the ECL properties of Ru(bpy)_3_^2+^. This advancement provides the first evidence for using f-CQDs in detecting biopharmaceuticals, highlighting their potential in analytical applications [50].

Carbon-rich precursors such as PW can be upcycled into various carbon compounds through processes such as pyrolysis. Not only does this conversion mitigate plastic pollution, but it also contributes to carbon sequestration on a global scale. The resulting carbon nanomaterials have widespread applications across diverse fields, including environmental remediation [51], electronics [52], energy storage [53], catalysis [54], biomedical engineering [55], and analytical sciences [56]. Recent inventories show that carbon-based nanomaterials are the second most commonly used class in consumer products, surpassed only by metal nanoparticles [57]. Conducting national research on PW waste can create economic opportunities, improve resource management, and advance technological capabilities. Furthermore, local research initiatives can offer practical solutions for PW management, raise public awareness about plastic pollution, and support local industries, fostering job creation and economic growth [58,59]. For example, the energy storage sector can utilize upcycled carbon-based nanomaterials like graphene and porous carbons as high-performance electrode materials in supercapacitors and lithium-ion batteries, essential for renewable energy systems and electric vehicles. In environmental remediation, industries can adopt porous carbon and carbon nanodots (CNDs) for water purification and air filtration, offering sustainable alternatives to traditional materials. The healthcare sector could leverage fluorescent carbon nanomaterials for biosensing, diagnostics, and enable advanced technologies in detecting biomarkers. Similarly, advanced manufacturing in electronics and semiconductors could integrate graphene and CNTs to enhance product performance while reducing reliance on imported raw materials. Additionally, upcycled nanomaterials can be used in agriculture as components of smart fertilizers and soil conditioners, promoting sustainable farming practices. These applications not only provide economic value but also foster innovation and generate employment opportunities in emerging sectors.

This review offers detailed insights into converting PW into carbonaceous compounds using ideal synthetic procedures such as thermal decomposition, CVD, FJH, and stepwise conversion by elucidating the transformation technologies of waste-to-value nanostructured materials and their potential applications. Overall, this review is a comprehensive resource for researchers and practitioners interested in understanding the latest developments in PW conversion and the multifaceted applications of resulting carbonaceous materials.

## 2. Previous Results

Various innovative methods have been demonstrated for transforming PW into valuable carbon-based materials, marking significant progress in the field. For instance, one study [60] provided an in-depth review of the catalytic upgrading of PET waste into a wide array of valuable products, including materials, fuels, and chemicals, by exploring techniques such as thermocatalysis, electrocatalysis, and photocatalysis. Electrocatalytic applications were also thoroughly examined [61], offering insights into the role of chemical catalysis in the conversion process and the development of PET-based electrode materials for energy conversion and storage [60].

A systematic review on converting PW into valuable 2D graphene-based materials [62] analyzed the types and conditions of plastics used as feedstock for synthesizing carbon materials, drawing data from 142 studies. The review found that 25% of the studies employed pristine plastics (Figure 2A), with PP, PET, PE, PS, and PVC being the most frequently used types (Figure 2A). These results reflect the current market demand for plastics [19].

The first documented conversion of PW into carbon materials was conducted in 2004 [63]. In their research, activated carbon was produced from PET waste through pyrolysis in an inert atmosphere, followed by CO_2_ activation, aiming to develop materials with high hydrogen adsorption capacity. Since then, numerous studies have explored the production of various carbon-based materials from plastics, including CNTs, carbon microspheres, 2D graphene materials, carbon nanofibers (CNFs), graphite, and fullerene. The increasing research focus on converting PW into carbon materials is shown in Figure 2B, illustrating a steady rise in studies since 2004. Nearly half of these studies focus on synthesizing CNTs, followed by activated carbon and 2D graphene materials (Figure 2C). The “other” category in Figure 2 includes composite materials, typically made by combining different carbon materials or blending them with other components like metals, polymers, or fibers.

Most research on converting PW into carbon materials is focused on optimizing synthesis conditions, comprising 53.5% of the total studies (Figure 2D). Environmental applications represent 22.6% of the research, while electrochemical and energy-related applications account for 21.2%. Within environmental applications, 73.5% are dedicated to developing adsorbent materials for water treatment. Many studies focus on electrochemical and energy applications, and 50% of them concentrate on creating materials for supercapacitors (Figure 2D).

### 2.1. General Impact and a Sustainable View of PW

Plastic, known for its versatility and durability, has become ubiquitous in modern life. However, the extensive and often careless use of plastic has resulted in a significant increase in PW, leading to serious environmental issues [64]. Specifically, PET in the form of microplastics poses a substantial risk to both terrestrial and aquatic ecosystems. Improper disposal of PW contaminates soil and water, adversely affecting plant and animal life [65]. Additionally, plastic debris can remain in the environment for centuries, releasing harmful chemicals and disrupting natural processes [66,67].

#### 2.1.1. General Impact

Since the development of early polymers like Bakelite and celluloid, plastic pollution has been a concern. Thermoplastic PET, which has been produced since 1940, is widely used in bottle packaging and textiles [68]. Figure 3 shows the diverse uses of plastics and the associated waste generated from PET production [69]. Global thermoplastic production is projected to reach 445.25 million tons, with an estimated 1200 million tons of plastic expected to accumulate in the environment or landfills by 2050. A considerable amount of PW is being released into the environment globally, contributing to the ongoing white pollution crisis. This type of pollution arises from the disposal of high molecular weight polymers such as PVC, PP, and PS, which are found in solid waste items like disposable tableware, packaging bags, plastic bottles, and agricultural mulch film. The various pathways through which PW is discharged into the environment are depicted in Figure 4 [70].

The seriousness of the PW issue is emphasized by the low rate of recycling compared to the high reliance on landfilling, which is often used for single-use plastics like packaging. Plastics with longer service lives, such as appliances and electronics, are discarded more slowly than single-use items such as bottles and bags. Since all plastics ever produced remain in the environment and contribute to plastic pollution for centuries, this issue persists. However, a study from 2016 found that the bacterium *Ideonella sakaiensis* can degrade PET by using it as a carbon source [71]. Further advancements include the development of an engineered PET depolymerase capable of recycling plastic bottles, which could be a starting point for creating nanostructured materials from waste PET. The degradability of plastics is influenced by factors such as shape, size, substituents, and molecular weight, with high molecular weight and strong chemical bonds making them more resistant to biodegradation [72]. Efforts are underway to foster a circular economy for PET plastics by designing products with recyclability in mind, enhancing recycling infrastructure, and promoting the use of recycled materials in new products.

#### 2.1.2. Sustainability Aspects

The increasing prevalence of plastics in the biosphere stresses the need for optimized technological solutions to break down synthetic polymers. Recent legislative measures, such as bans on single-use plastics, bottle return schemes, and recycling initiatives, aiming to address this issue [73] have been introduced recently [74]. The upcycling approach aligns with and enhances the effectiveness of these bans by offering an economically viable solution for repurposing single-use plastics, which are otherwise challenging to manage. Instead of merely restricting their use or relegating them to landfills, the proposed strategy converts single-use plastics into high-value carbon-based nanomaterials, such as graphene, CNTs, and porous carbons. This transformation not only reduces environmental pollution but also creates a market-driven incentive for collecting and processing these plastics. By integrating upcycling into waste management systems, these policies can achieve greater sustainability outcomes, reducing the reliance on bans alone and fostering a circular economy. For instance, India enacted a ban on single-use plastics in July 2022, promoting bamboo-based alternatives in its northeastern regions [75]. This initiative includes innovations like sugarcane straws and exchange kiosks offering cash, gift cards, and food in return for PW, benefiting local communities.

In the United States, the Department of Energy and Environmental Protection Agency are enhancing recycling efforts and reducing greenhouse gas emissions. The Plastics Innovation Challenge targets solutions for 90% of plastics, while the National Recycling Strategy aims for a 50% recycling rate by 2030 [76]. Spain is also taking steps by imposing fees on single-use plastic containers and landfill deposits, banning microplastics in cosmetics, and promoting a circular plastic economy focused on reducing waste and encouraging reuse and recycling [77].

Scientific and technological advancements could foster job creation, improve healthcare, manage industrial waste, and enhance environmental safety [78]. Specifically, upcycling PET plastics into carbonaceous materials presents a viable and sustainable method for addressing plastic pollution and aligning with the United Nations’ Sustainable Development Goals (SDGs). This approach supports clean water and sanitation (SDG 6), climate action (SDG 13), responsible consumption and production (SDG 12), and life below water (SDG 14) [79]. By integrating these sustainable practices into industrial processes, significant progress can be made toward achieving national and international environmental objectives.

### 2.2. Plastics from Waste to Worth

To address the issue of PW and reduce environmental pollution, incineration and landfilling are commonly employed. While incineration can reduce the volume of PW by 90–99% [80], it also produces harmful byproducts such as furans, hydrocarbons, polycyclic aromatic compounds, and carbon monoxide [81]. Moreover, the residual ash from incineration contains microplastics, which pose significant risks to ecosystems. These environmental concerns have accelerated research into more sustainable PW recycling methods, driving innovation in the field.

There are three main approaches to plastic recycling: mechanical, biological, and chemical. Each method breaks down PW through distinct processes to create new products [82]. Mechanical recycling involves sorting, washing, shredding, and re-granulating plastics to produce secondary materials [83]. Although widely used, this method often results in plastics with weaker mechanical properties and inconsistent flow behavior, limiting their use in industries like packaging and medicine [84]. Additionally, the process is labor- and energy-intensive. Biological recycling uses microorganisms or enzymes to break down plastics containing hydrolyzable ester or amide bonds, though this method is limited to specific types of plastics like polyethylene terephthalate (PET). Enzymes such as PETase can break down PET into its monomers under mild environmental conditions, making this approach energy-efficient and eco-friendly. However, its application is limited to a narrow range of plastics, as enzymes often lack the specificity or robustness to degrade more complex or inert materials, such as polyethylene (PE) and polypropylene (PP). Additionally, biological processes can be slow and are susceptible to inhibitors in mixed or contaminated waste streams, further limiting their applicability in real-world recycling settings.

Chemical recycling employs heat or chemical reactions to break plastics down into their monomers, which can be repolymerized into high-quality, virgin-grade materials. This method has gained attention due to its ability to restore PW to its original form without compromising quality or economic value. However, chemical recycling often requires high energy input and, in some cases, uses toxic solvents or catalysts, raising concerns about its environmental impact and economic feasibility. Recently, bio-cycling of chemically processed PW has surfaced as an innovative alternative to traditional plastic recycling methods. Combining biological and chemical methods for plastic recycling has the potential to address the limitations of current recycling technologies and significantly expand their applicability. Chemical processes can act as a pre-treatment step to break plastics into smaller, more accessible intermediates, which are then further degraded by enzymes or microorganisms. For example, pyrolysis or hydrolysis can reduce complex plastics into oligomers or other fragments that enzymes can efficiently process. Moreover, the chemical pre-treatment phase can help to reduce contaminants or inhibit agents that may hinder enzyme activity, enabling a more effective biological process under mild conditions. This hybrid approach reduces the energy demands and environmental impacts of conventional chemical recycling while extending the range of materials suitable for biological processing [85]. Upon combining the accessibility of chemical methods with the mild conditions of biological recycling, hybrid approaches hold the potential for large-scale, efficient recycling systems that could transform PW management.

As underlined in a recent study [69], four categories for plastic recycling terminology were defined: primary, secondary, tertiary, and quaternary recycling. Primary recycling involves the mechanical reprocessing of scrap plastic materials into products with properties equivalent to the original material. Secondary recycling refers to mechanical reprocessing where the resulting products have lower properties than the original. Tertiary recycling involves the recovery of valuable chemical constituents, such as monomers or additives, while quaternary recycling focuses on energy recovery from PW, as shown in Figure 5. Transforming waste into more valuable materials or products through recycling is termed “valorization”. Primary recycling generally encompasses the direct reuse of waste plastics, while tertiary and quaternary recycling fall under valorization processes [86,87,88].

Additionally, research is increasingly focused on integrating chemical, biological, and hybrid catalytic techniques to enhance plastic breakdown, recycling, and upcycling efforts, which are crucial for addressing the global PW crisis [89]. For example, researchers developed a hybrid catalytic process for upcycling PET involving chemical and biological catalysis [90]. In this study, PET is first chemically depolymerized via glycolysis using ethylene glycol (EG) and titanium butoxide as a catalyst, yielding bis(2-hydroxyethyl) terephthalate (BHET). This intermediate, more accessible for microbial breakdown, then undergoes biological catalysis through an engineered strain of *Pseudomonas putida* KT2440. This engineered *Pseudomonas putida* strain expresses PETase and MHETase enzymes, which hydrolyze BHET into terephthalic acid (TPA) and ethylene glycol (EG). Through further metabolic engineering, the *P. putida* strain was modified to catabolize TPA to protocatechuate (PCA) and convert PCA into β-ketoadipic acid (βKA), a high-value precursor for performance-advantaged nylon. In bioreactor tests, this process achieved βKA production at 15.1 g/L, with a molar yield of 76%, demonstrating the feasibility and efficiency of hybrid catalytic upcycling of PET into high-value biochemicals. Furthermore, the economic analysis suggests that this process could be scalable to a pilot or industrial level, with enzyme production costs estimated at $1.25 per kilogram of protein according to [USDA National Nutrient Database, (USDA)] [91]. This indicates that the upcycling process might achieve a production cost equivalent to only 4% of the ton price of virgin PET, highlighting its potential for cost-effective, large-scale application. When traditional recycling methods prove costly or impractical, upcycling offers a promising alternative. This innovative approach leverages chemical and engineering strategies to transform PW into valuable raw materials for synthesizing new products [92]. The key challenge lies in converting PW into materials with higher economic value, a complex issue involving chemical, economic, and environmental factors. Emerging concepts aim to unlock high-value markets in the circular economy by repurposing PW into more useful and valuable materials.

## 3. Overview of PW-Derived Carbon-Based Nanomaterials Applications

A promising approach to addressing PW involves upcycling it into valuable carbon-based nanomaterials. This strategy not only mitigates plastic pollution but also contributes to reducing carbon emissions. Carbon-based nanomaterials, including fullerene, flash graphene, carbon spheres, porous carbon, and carbon nanosheets, have garnered significant attention for their versatile applications across various fields such as environmental remediation, energy storage, biomedicine, and catalysis. Regarding the environmental applications, their large surface area and tunable properties make them highly effective sorbents for removing heavy metals like lead, mercury, and arsenic from water. Furthermore, one of the most significant benefits of using these materials for heavy metal adsorption is their regeneration ability, which enhances their environmental sustainability for long-term applications. Carbon-based nanomaterials derived from PW can undergo multiple regeneration cycles with minimal reduction in adsorption efficiency, reducing the need for frequent replacement. Various regeneration techniques, including acidic or alkaline desorption and thermal treatments, effectively restore the active sites by removing adsorbed metals without degrading the core carbon structure. This allows the materials to retain their adsorption performance across cycles, providing reliable and cost-effective treatment solutions [93]. In energy storage, materials like graphene and porous carbon are key components in supercapacitors and lithium-ion batteries, enhancing energy density, charge/discharge rates, and overall efficiency. In biomedicine, fullerenes and carbon quantum dots are utilized in bioimaging and as drug delivery systems, owing to their biocompatibility and ability to function with therapeutic agents for targeted treatments. Carbon nanomaterials are also integral to developing high-efficiency solar cells, such as organic and perovskite cells, where they improve electron mobility and device stability. Additionally, their catalytic properties, particularly in hydrogen evolution and CO_2_ reduction reactions, make them valuable in industrial and environmental processes. These nanomaterials are also used in water purification, where their ability to adsorb contaminants or act as filtration membranes enhances water treatment efficiency. Overall, the unique mechanical, electrical, and chemical properties of carbon-based nanomaterials position them as critical components in a wide range of innovative technologies (Figure 6) [92,94,95,96,97].

The continued research and development of carbon-based nanomaterials from PW holds the potential to revolutionize technology, tackle global challenges, and drive sustainability initiatives. By transforming PW into valuable resources, this approach offers a promising solution to the environmental impacts of plastic pollution, while also opening opportunities for innovation and circular economic growth. Adopting upcycling as a core strategy in waste management could be a significant step toward a more sustainable and eco-friendly future.

### Adsorption of Pollutants

Adsorption is an effective method for removing pollutants from contaminated water due to its high efficiency, cost-effectiveness, ease of implementation, and adaptability [98]. The success of this approach depends largely on the physicochemical properties of the adsorbent, including its nanostructure, surface functional groups, specific surface area (SSA), and wettability, as well as the characteristics of the pollutants being targeted. This section explores the application of carbon-based nanomaterials derived from PW(PWCMs) for the adsorption of organic pollutants, such as dyes and antibiotics, and inorganic contaminants as well.

Organic Pollutants: Carbon-based materials derived from PW, including porous carbon [99], CNTs [100], carbon nanosheets [101], and graphene [102], have been employed as high-performance adsorbents for organic pollutants. These materials remove contaminants through electrostatic attraction, pore filling, hydrogen bonding, and π–π interactions (Figure 7A) [103]. To enhance their performance, strategies have focused on optimizing porosity and composition. For example, Li, Chen [103] synthesized porous carbon by co-pyrolyzing waste polypropylene (PP) with cyanobacteria, achieving a methylene blue (MB) adsorption capacity of 667 mg/g. The pyrolysis enhanced the carbon pore structure and introduced functional groups like C=O, O–H, and N–H, which significantly improved adsorption efficiency.

Another effective approach is KOH activation, which enhances the porosity of PWCMs [101]. The use of solid KOH during activation has been shown to produce heterogeneous porosity, including macropores (1.77 cm^3^/g), mesopores (0.81 cm^3^/g), and micropores (0.71 cm^3^/g). This technique also resulted in a high specific surface area of 1990 m^2^/g, leading to superior pollutant removal capacity [104]. These advancements demonstrate the potential of PW-derived carbon materials for addressing water contamination challenges.

Combining PWCMs with functional nanomaterials, such as metal oxides or alginates, significantly improves their adsorption capacity [105]. These additives introduce new active metal sites, enabling the adsorption of organic pollutants through chemical bonding [106]. For instance, Rai and Singh [107] demonstrated that Fe–O groups in Fe_3_O_4_ served as nucleation sites, enhancing the adsorption of cephalexin onto magnetic PET-based activated carbon. Additionally, magnetic PWCM–metal oxide composites offer the practical advantage of easy separation from the reaction solution, simplifying the recycling and reuse of the adsorbent material.

Inorganic Pollutants: PWCMs have been effectively utilized to remove inorganic contaminants, particularly heavy metal ions [108]. Oxygen-containing functional groups, such as C=O and O–H, serve as the primary adsorption sites for these pollutants [109]. Xu, Zhu [110] demonstrated that hydrochar derived from PVC containing –SO_3_H groups exhibited enhanced adsorption capacity for Cu(II) and Cr(VI) ions (Figure 7B). The removal of Cu(II) ions was primarily driven by electrostatic interactions, whereas Cr(VI) anions were complexed with phenolic –OH groups and subsequently reduced to Cr(III) ions through interactions with C=C groups on the hydrochar surface. Additionally, a polypyrrole-modified PET-derived carbon composite proved effective for the removal of NO_3_^−^ ions, utilizing ion exchange and electrostatic attraction mechanisms [111].

**Figure 7 polymers-17-00063-f007:**
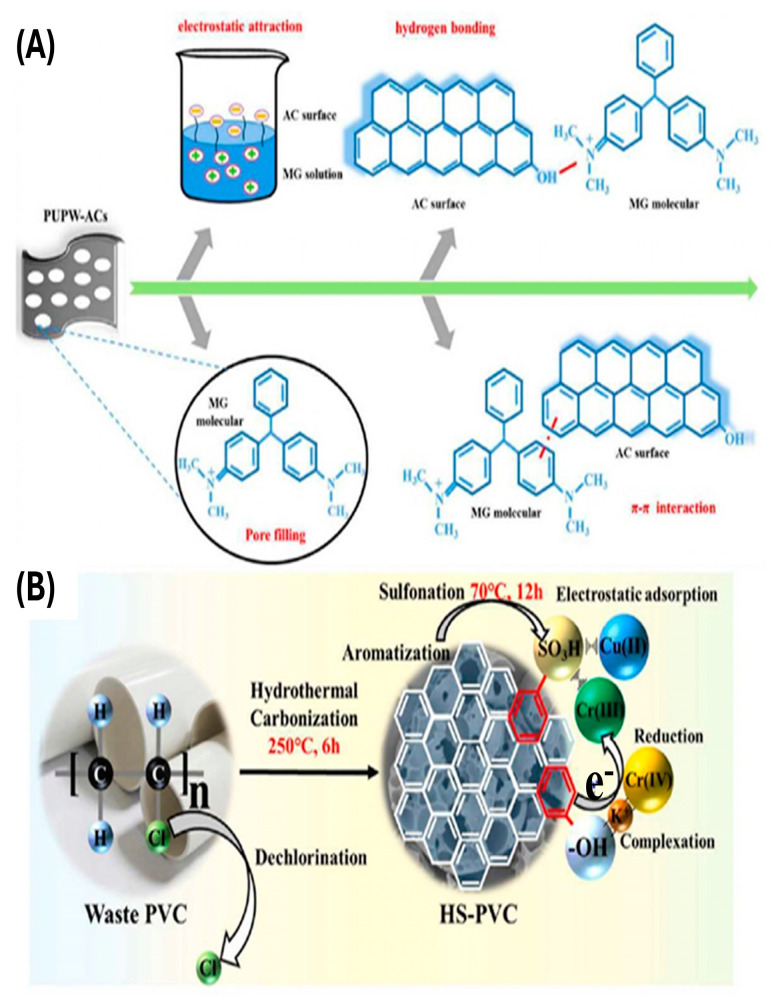
(**A**) Adsorption of MG (malachite green) dye on AC derived from polyurethane waste. Reprinted from [103] with permission from Elsevier, Copyright (2020). (**B**) Summarizing the adsorption removal of Cu (II), and Cr (VI) via PVC-derived hydrochar. Reprinted from [110] with permission from Elsevier, Copyright (2020).

## 4. Carbon-Based Materials Produced from PW Conversion

Repurposing PW to create value-added products and materials has become a promising strategy to enhance the appeal of plastic recycling [112]. Given the high carbon content of plastics, converting them into valuable carbon-based materials is an effective approach for both waste management and carbon material production [62]. Various techniques have been developed to produce high-quality carbon materials such as graphite, graphene, fullerene, activated carbon, and CNTs, which have applications in numerous fields [113,114]. This section will explore the main methods used to convert plastic into valuable carbon materials, including thermal decomposition, FJH, CVD, and stepwise conversion, as illustrated in Figure 8. Each method will be examined in detail in the following subsections.

While traditional graphene synthesis often depends on fossil fuels, researchers are investigating PW as a precursor for producing graphene. PW can be converted into graphene sheets through different methods such as chemical exfoliation, offering a sustainable alternative for this advanced material. For example, one study demonstrated that waste plastic bottles, drink containers, and yogurt cups, when subjected to catalytic pyrolysis at 90 °C, produced high-quality graphene with reduced oxygen content and a surface area of 54.872 m^2^/g. Another innovative approach involved upcycling waste PET into a nanocomposite by first growing laser-burned UiO-66 crystals. This graphene-like material, integrated with the waste plastic matrix, showed high mechanical stability and surface conductivity of 10.4 ± 3.1 Ω/square [115]. Graphene’s potential applications are extensive, spanning electronics, energy storage, sensors, medical devices, and water treatment [116,117,118]. Using PET waste for graphene production supports the principles of a circular economy and reduces the environmental impact associated with traditional graphene production methods [119]. However, producing graphene from PW via chemical exfoliation is challenging, particularly in addressing impurities and structural defects. PW is inherently heterogeneous, containing contaminants such as additives, stabilizers, dyes, and fillers, which can interfere with the exfoliation process and lead to non-uniform graphene production. Residual impurities like food, oils, and dirt further compromise the reaction environment, while mixed polymer types in waste streams add to the difficulty of achieving consistent precursor quality [120].

Structural defects in the resulting graphene are another concern. During the exfoliation process, heteroatoms such as oxygen, nitrogen, or chlorine, originating from degraded plastics or additives, may become incorporated into the graphene lattice. This results in defects that affect the material’s properties. Furthermore, the variability of the plastic feedstock often leads to graphene flakes with inconsistent sizes and thicknesses. In some cases, the process may produce significant amounts of amorphous carbon or other byproducts instead of high-quality graphene, reducing the material’s usability [121].

To mitigate these issues, pre-treatment of PW is essential. Techniques such as sorting, cleaning, and thermal or chemical decomposition help reduce contaminants and simplify the carbon source. Additionally, process optimization, including the careful selection of reagents and control of reaction conditions, can minimize defect formation. Post-production purification methods, such as filtering, centrifugation, or annealing, can further enhance graphene quality. Using homogeneous plastic feedstock, like isolating specific polymers such as polyethylene, can also reduce variability and improve process reliability [122].

Despite these challenges, producing graphene from PW offers a promising pathway for recycling and sustainable material development. Addressing impurities and structural defects through meticulous feedstock preparation, process control, and purification techniques is critical to realizing high-quality graphene for practical applications.

Researchers have been exploring inventive methods to repurpose PW into valuable carbon materials, each offering unique properties and numerous benefits, as illustrated in Figure 8. These PW-derived carbon materials exhibit distinctive attributes that enhance their suitability for several applications. Table 1 provides a comprehensive summary of the techniques used for plastic recovery, including details on operational conditions, carbon structures, yields, and potential applications.

### 4.1. Thermal Decomposition

Thermal decomposition is a chemical process that occurs when a substance is exposed to elevated temperatures ranging from 400 to 600 °C, causing it to break down into simpler components. According to the American Society for Testing and Materials (ASTM), this process involves significant chemical alterations because of the high heat exposure [139]. This differs from thermal degradation, which refers to the deterioration of physical, mechanical, or electrical properties when materials are subjected to high temperatures. During thermal decomposition, solid substances release flammable vapors that can ignite above the material [140]. This reaction can sustain itself as long as the heat from the burning gases is sufficient to maintain the production of these vapors, potentially creating a feedback loop where the material continues to burn, ensuring consistent product quality.

Pyrolysis is a highly efficient thermal decomposition process for managing PW, enabling the breakdown of diverse plastic types such as PS, PP, PE, HDPE, LDPE, and mixed plastics in an oxygen-free environment. This process yields valuable chemicals in the form of pyrolytic gases and oils, alongside heat energy. PW pyrolysis can produce hydrogen Fuel, diesel that can be used to run engines for industry and transport, and crude oil that can be refined into gasoline. Products from pyrolysis are frequently used as cogeneration oil, engine fuel, solid fuel, and raw materials for additional processing to make polymers and other goods [141].

One of the main advantages of pyrolysis compared to other recycling methods is its flexibility, as it eliminates the need for separating plastics based on polymer type, making the process more convenient and cost-effective. The process can be carried out using different reactors and heating sources, such as electrical heaters or burners, with the choice of reactor and operating conditions significantly impacting the quality and yield of the products. Higher temperatures and longer residence times typically result in increased oil production, while lower temperatures and shorter residence times favor the production of gases. One challenge associated with the process is the low thermal conductivity of plastics, which can lead to uneven heating, localized overheating, and excessive carbon buildup on reactor surfaces. Table 2 provides an overview of various carbonization processes, and the conditions required for synthesizing carbon materials [142]. Thermal decomposition, particularly pyrolysis, is an efficient process for converting PW into valuable products like carbon-based nanomaterials, syngas, and liquid hydrocarbons. Operating at temperatures between 400 °C and 900 °C in an oxygen-free environment, it stands out due to its ability to process mixed and contaminated PW without requiring energy-intensive preprocessing [143], compared to other methods such as CVD. CVD, while producing high-quality graphene, demands the conversion of plastic into gaseous precursors and operates at much higher temperatures (~1000 °C), making it significantly more energy-intensive and less practical for PW. Similarly, stepwise conversion methods are less energy-efficient due to multiple stages that increase energy demands and operational complexity. FJH is an emerging technique that is energy efficient and enables rapid graphene production within milliseconds. This method minimizes energy losses due to its short reaction time and offers high-quality graphene production. However, FJH requires specialized equipment and precise operational controls, limiting its scalability to smaller applications despite its energy efficiency [144]. Pyrolysis, on the other hand, benefits from energy recovery systems, such as utilizing the syngas produced during the process as a fuel source, thereby reducing energy requirements. Moreover, advanced reactor designs like fluidized or rotary kilns improve energy efficiency by ensuring even temperature distribution and reducing heat losses [145].

Thermal decomposition is an efficient and cost-effective method for producing advanced carbon nanomaterials, such as 3D sponge-like nitrogen-doped graphene (NG), from waste PET bottles. By combining discarded PET with urea at varying temperatures, this straightforward and eco-friendly one-step process enables the synthesis of NG with unique properties. This approach not only offers a sustainable solution for managing PET waste but also facilitates the production of high-value carbon materials with potential applications in various fields [148].

In this process, PET waste was mixed with urea in varying ratios and placed in a stainless-steel autoclave within an electric furnace. The mixture underwent heating for five hours at temperatures of either 600 °C or 800 °C, followed by gradual cooling overnight. The resulting dark product was ground into nitrogen-doped graphene (NG), where nitrogen atoms were incorporated into the graphitic lattice (the atomic percentage of nitrogen in each NG sample and its components). For comparison, both pure graphene (G) and NG samples with different urea-to-PET ratios were synthesized. X-ray photoelectron spectroscopy (XPS) analysis indicated that reducing the urea content led to a corresponding decrease in nitrogen levels in the samples. Electrochemical assessments using cyclic voltammetry and impedance spectroscopy demonstrated that nitrogen incorporation improved charge transfer and ion diffusion. Transmission electron microscopy (TEM) images revealed large, few-layered graphene structures with nitrogen atoms embedded within a polycrystalline framework, while high-resolution TEM confirmed corrugated microstructures (Figure 9A–C). Selected area electron diffraction (SAED) showcased wrinkled features in the material (Figure 9D). The thermal decomposition of urea and PET generated gases that formed a porous structure, optimizing ion pathways and surface area for electrochemical energy storage. This method is not only cost-effective and environmentally sustainable but also holds great potential for scaling up nitrogen-doped graphene production from PW, making it a promising solution for upcycling plastic.

The production of carbon nanomaterials (CNMs) and hydrogen-rich gases as byproducts was effectively achieved through pyrolysis and catalysis using various types of PW, as depicted in Figure 10 [149]. Two sets of catalysts were synthesized: one via the conventional impregnation method, involving Co, Fe, and Ni as active metals supported on MgO (Co-Fe/MgO, Co-Ni/MgO, and Fe-Ni/MgO), and the other via the coprecipitation method (Co-Fe-Mg, Co-Ni-Mg, and Fe-Ni-Mg). The experiment was carried out using a two-stage reactor. In the first stage, PW underwent pyrolysis to produce hydrocarbon volatiles, and in the second stage, these volatiles were reformed catalytically. The performance of each bimetallic catalyst was evaluated by analyzing the resulting carbon nanomaterials (CNMs) and valuable gases. The study also compared the efficiency of converting four different types of plastics: LDPE plastic bags, PP plastic bottles, PS plastic lids, and PET water bottles. The comparison was based on product yield, carbon purity, and adsorption capacity. Notably, the Fe-Ni-Mg catalyst synthesized via the precipitation method at an increased pH showed the highest efficiency in converting PW. Among the plastic types, LDPE plastic bags produced the highest hydrogen yield at 35.27 mmol/g of plastic, while PS plastic lids generated the largest amount of CNMs at 38.26 wt%. The CNMs exhibited excellent adsorption properties, with an adsorption capacity of approximately 180 mg/g CNM for metal cations like Fe, Ag, and Ni, demonstrating their potential use in wastewater treatment.

### 4.2. Chemical Vapor Deposition (CVD)

CVD is a sophisticated technique widely used to produce high-quality solid materials, particularly in controlled vacuum environments. This process operates by initiating chemical reactions of organometallic or halide compounds, resulting in the formation of the desired material. One of the significant applications of CVD is transforming PW into CNTs. The CVD process generally consists of two main stages: initially, PW is broken down into volatile vapors at moderate temperatures in an oxygenated environment; in the subsequent stage, these vapors are converted into CNTs under high temperatures and pressures, facilitated by a catalyst. Figure 11 illustrates the specific steps of the CVD process.

However, while this method demonstrates a high growth rate and substantial yield of CNTs, several scalability challenges remain, particularly concerning feedstock variability, energy consumption, and catalyst performance. PW consists of diverse polymers that decompose into different volatile compounds during pyrolysis. This heterogeneity can lead to inconsistent CNT quality and yield. Researchers are addressing this by optimizing catalyst design, such as using bimetallic systems like FeMo/MgO, which have proven effective at handling mixed hydrocarbon streams, resulting in high-purity CNTs. These catalysts have been shown to improve yield and purity, with some processes achieving carbon yields exceeding 500% and CNT purities more than 97%. Moreover, these catalysts offer superior performance by stabilizing the metal particles and preventing sintering at high temperatures [150]. Energy consumption in CVD remains a significant challenge due to the high operational temperatures, typically above 600 °C, increasing the energy demands of the process, and presenting a barrier to large-scale production. Recent innovations in reactor design and energy recovery systems have been developed to reduce thermal losses, making the process more efficient [151]. Some studies have even integrated pyrolysis and CVD into a single process, minimizing energy consumption while maintaining high CNT yields. This combined approach not only enhances energy efficiency but also streamlines production, making large-scale CNT synthesis from PW more viable [152]. Current research in this area aims to optimize key parameters of the process to scale up the use of PW for producing CNTs, graphene, and other advanced carbon materials. Despite CVD advantages, such as high growth rates and substantial yields, the CVD process remains complex and energy-intensive. However, ongoing innovations in catalyst design, process integration, and system optimization pave the way for scalable and sustainable CNT, graphene, and other advanced carbon materials production.

The recycling of PW into valuable materials was achieved using a cost-effective, simple, and dependable CVD system. A recent economic evaluation study compared the upcycling CVD process with the traditional method of producing CNTs. The cost of the upcycling process was 2999 $ kg^−1^ which was slightly higher than the conventional one of 2930 $ kg^−1^ but a 20% reduction of the argon flow rate made the upcycling technique surpass the conventional process [153]. This approach enables the industrial-scale upcycling of various types of PW, such as PET, PE, poly(methyl methacrylate) (PMMA), PP, PS, and PVC, into high-quality graphene foil (GF), as depicted in Figure 9. This process demonstrates the potential for large-scale transformation of common plastic materials into advanced carbon products, contributing to both waste reduction and material innovation. Polymers and PW from daily activities were selected as carbon sources for GF synthesis (Figure 12A), labeled from 1 to 7 based on the substrate types. To maintain a continuous carbon source supply and ensure thorough interaction with the nickel foil catalyst, the materials were gradually introduced into the heated zone (Figure 12B). The growth process occurred at 1050 °C using a 40 μm nickel foil as the catalyst. Large-scale production of paper-like GF was achieved using PS as the starting material (Figure 12C). The resulting GF displayed a hierarchical, uniform laminated structure composed of multi-layer graphene with good crystallization (Figure 12D–F). The method demonstrated similar structural outcomes when other plastics were used, showcasing its feasibility and versatility.

A polycrystalline nickel foil was used as a catalyst and placed in a quartz tube, then heated to 1050 °C for 30 min. in an Ar/H_2_ atmosphere to remove surface oxides. Following annealing, the carbon source was melted and gradually injected into the CVD reactor. After the process, graphene film (GF) was obtained by etching away the nickel foil using a ferric chloride/hydrochloric acid solution, followed by washing with deionized water. The resulting GF displayed exceptional electrical conductivity of 3824 S.cm^−1^, which surpasses that of typical graphene films processed at ultrahigh temperatures (2200–2500 °C). Additionally, GF can be employed as flexible components, such as standalone electrodes in foldable Li-ion batteries, showcasing stable electrochemical performance. It also demonstrated rapid and low-voltage reactivity, functioning as a flexible electrothermal heater that can reach 322.6 °C with just a 5 V input [154,155]. The efficient upcycling of PW into valuable GF opens new possibilities for graphene utilization across diverse fields.

**Figure 11 polymers-17-00063-f011:**
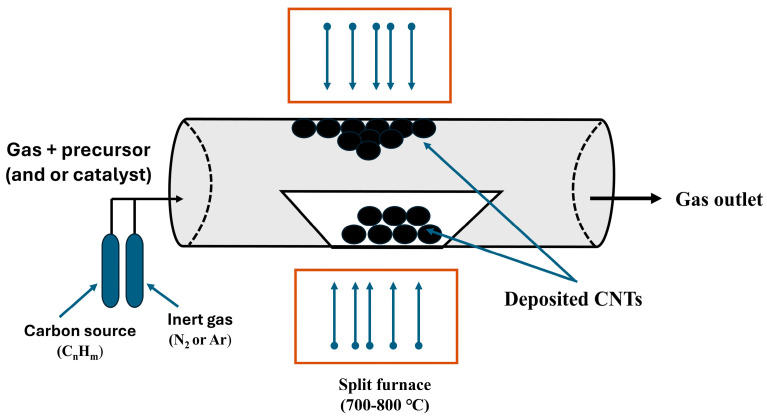
Schematic representation of CVD setup showing main components and stages.

**Figure 12 polymers-17-00063-f012:**
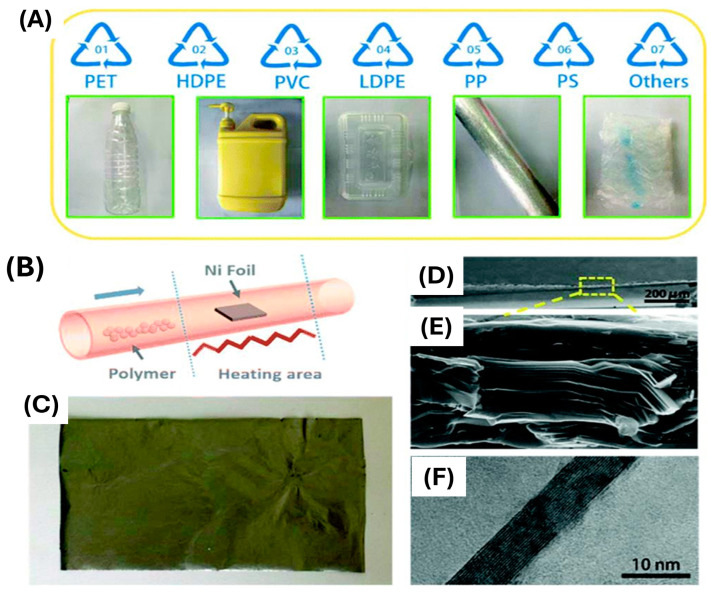
From trash to treasure: converting PW into valuable graphene foil (GF). (**A**) Images of the raw waste plastic materials utilized in the experimental process, showcasing the types and conditions of the input materials. (**B**) Schematic representation of the fabrication process for the GF, outlining the key steps involved from PW to final product. (**C**) Photograph of the produced graphene foil, highlighting its physical characteristics. (**D**,**E**) Scanning electron microscopy (SEM) images of the graphene foil’s edge at different magnifications, with (**D**) showing a scale of 200 µm and (**E**) presenting a zoomed-in view at 500 nm, illustrating the structural details and surface morphology (**F**) Transmission electron microscopy (TEM) image depicting the few-layer graphene within the structure of the GF, providing insights into its layering and crystallinity. Reproduced from [156] with permission from Royal Society of Chemistry, Copyright (2017).

A recent study presented an innovative approach to upcycling discarded PW into high-quality monocrystalline graphene on polycrystalline copper foil. This was accomplished through ambient pressure CVD (AP-CVD), where copper foil served as the substrate, and PW acted as the carbon source (Figure 13) [157]. To reduce nucleation sites and facilitate graphene growth, the copper was annealed in a hydrogen atmosphere at 1020 °C for 30 min before introducing the carbon feedstock. The graphene crystals were subsequently transferred onto a SiO_2_/Si substrate using a PMMA layer, which was later dissolved with acetone. A horizontal atmospheric pressure CVD (AP-CVD) system, featuring a 90 cm long quartz tube with a 50 mm diameter placed in two split furnaces, was used for graphene synthesis (Figure 13A). The substrate was 20 μm thick, 99.99% pure copper foil (Ni-laco Corp.). Solid waste plastic, typically used for material packaging, served as the carbon source (Figure 13B). Optical microscopy was employed to examine the growth and morphology of graphene crystals on the copper foil. To enhance the visibility of graphene domains, selective oxidation of the copper surface was conducted by annealing at 150 °C for 10 min under atmospheric conditions. This low-temperature oxidation did not affect the pristine quality of the graphene, but it converted the copper foil to copper oxide, creating a color contrast that made the graphene domains visible.

Figure 13C shows Raman spectra of the graphene crystal, demonstrating its crystalline nature, layer structure, and defects. Raman spectra, taken from four different points on the crystal, reveal a minimal defect-induced D band, indicating the high quality of the graphene. Graphitic G peaks and second-order 2D Raman peaks appear at 1590 and 2700 cm^−1^, respectively. The intensity ratio of the G to 2D peak confirms the presence of single-layer graphene, similar to the hexagonal graphene structure. The full width at half maximum (FWHM) values for the G and 2D peaks are 17 cm^−1^ and 32 cm^−1^, respectively. These Raman results demonstrate the high quality of graphene crystals, irrespective of their hexagonal or round shapes.

Figure 13D also displays the Raman spectra of a graphene crystal, highlighting the variation in the number of layers as inferred from color contrast. Notably, Raman spectra were taken at different points (1 and 2) on the graphene crystal. The rate at which the decomposed polymer was injected, derived from the pyrolyzed PW, played a crucial role in controlling the growth of individual graphene crystals. By carefully regulating this rate, the researchers successfully produced large hexagonal and round-shaped graphene structures.

### 4.3. Stepwise Conversion

The stepwise conversion method is an adaptable and efficient process designed to transform PW into valuable carbon materials through a multi-stage approach. Initially developed for industrial applications, this method offers several advantages, such as greater control and enhanced efficiency compared to other techniques. The process begins with the pyrolysis of PW, where the material is heated in an oxygen-free environment, resulting in the breakdown of plastic into gaseous hydrocarbons and solid residues. In the subsequent stage, these gaseous products are directed to interact with specifically selected catalysts. The role of these catalysts is crucial, as they drive the chemical reactions necessary to convert the hydrocarbons into carbon-based materials like CNTs. By using tailored catalysts, this stepwise conversion process can achieve targeted chemical transformations, yielding high-quality carbon materials with versatile applications. Compared to single-step methods such as thermal decomposition or FJH, the stepwise approach offers unique advantages and presents specific challenges. Several benefits distinguish the stepwise method from its single-step counterparts. First, the separation of decomposition and synthesis stages allows for fine-tuning of reaction conditions, resulting in carbon materials with specific pore structures or dimensions, which is particularly advantageous for CNT production. Additionally, the use of targeted catalysts ensures efficient utilization of hydrocarbons, leading to higher yields and better material quality. The method also incorporates energy recovery from exothermic reactions, enhancing overall sustainability and reducing energy consumption. Despite its advantages, the stepwise conversion method is not without challenges. The increased complexity of managing multiple reaction stages raises operational costs and energy requirements compared to simpler methods like thermal decomposition and FJH. Furthermore, achieving consistent catalyst performance at industrial scales remains a significant hurdle. These challenges must be addressed through advancements in catalyst design, process optimization, and automation to make the method more feasible for large-scale adoption. Overall, by overcoming these challenges through scientific and technological advancements, the stepwise conversion method not only provides a sustainable strategy for managing PW but also upcycling waste into high-value products, demonstrating its potential in both environmental remediation and advanced material production [27].

In a previous study, multi-walled CNTs (MWCNTs) were successfully synthesized from PE waste using a stepwise pyrolysis–combustion technique, as demonstrated in Figure 14A [158]. The process begins with the pyrolysis of PE in forms like strips, pellets, or ground particles, which generate gaseous byproducts such as light hydrocarbons (LHCs) and hydrogen (H_2_). These vaporized products are then mixed with oxygen-containing gas and ignited, producing a fuel-rich flame. This flame is considered the key to the subsequent stage, where MWCNTs are synthesized. The synthesis involves separating the pyrolysis and combustion stages into distinct reactors. This separation allows the precise control of the process conditions, ensuring optimal CNT formation. The effluent from the combustion process provides the necessary feedstock for CNT synthesis, yielding approximately 10% of the mass of the initial carbon content before purification. In an industrial setting, this process could be made more energy-efficient by recovering the exothermic energy from PW combustion to offset the energy demands of the pyrolysis stage, thereby reducing production costs for CNTs.

SEM images taken at various magnifications, without any prior sample treatment, reveal the formation of tubular structures on the substrate surfaces (Figure 14B,C). The CNTs shown were produced through the combustion of post-consumer HDPE pyrolysis at two temperatures, 800 °C and 750 °C, with a 50% oxygen mole fraction at the venturi; the substrate/catalyst used was 304 stainless steel. 

In a different study, a two-stage process was utilized to PP into multi-walled CNTs (MWCNTs) and hydrogen-rich gas [159]. The method consists of two primary phases: first, catalytic pyrolysis of PP is conducted using HZSM-5 zeolite, followed by the catalytic breakdown of the resulting pyrolysis gases to form MWCNTs and hydrogen, with nickel catalysts in a moving bed reactor. Scanning electron microscopy (SEM) and transmission electron microscopy (TEM) confirmed the presence of MWCNTs in the carbon products, while X-ray diffraction (XRD) and thermogravimetric analysis (TGA) showed that higher decomposition temperatures resulted in CNTs with better crystallinity. The study examined how pyrolysis temperatures (ranging from 550 °C to 750 °C) and decomposition temperatures (from 500 °C to 800 °C) impacted the process efficiency. It was found that both MWCNT production and hydrogen concentration reached their peak at a decomposition temperature of 700 °C. Furthermore, MWCNTs generated at higher temperatures displayed improved graphitization and thermal stability. The pyrolysis temperature also influenced both the gas composition and the MWCNT yield. This research presents a viable method for transforming PW into valuable carbon nanomaterials and hydrogen.

The results from these studies suggest that pyrolysis temperature is a crucial factor in determining both the amount and composition of the pyrolysis gas, as well as the overall product yield in the stepwise conversion process. Furthermore, the selection of catalysts greatly influences the properties of the resulting carbon materials, leading to variations in their forms, such as straight, twisted, or coiled structures. Although the pyrolysis and decomposition stages generally require high temperatures and significant energy consumption, the process’s energy efficiency can be enhanced by incorporating a heat exchanger to recycle the heat generated back into the pyrolyzer. This heat recovery system assists in the gasification of the precursor material, making the process more energy-efficient, sustainable, and cost-effective.

### 4.4. Flash Joule Heating (FJH)

FJH is an advanced technique for producing high-quality carbon materials, particularly for generating high-performance carbon nanomaterials. Through FJH, waste plastic can be transformed into large quantities of high-grade turbostratic graphene by discharging direct current through the precursor materials via large capacitors. There are two forms of FJH used to create turbostratic flash graphene (FG): direct current FJH (DC-FJH) and alternating current FJH (AC-FJH). This cutting-edge method leverages electrical energy to trigger rapid joule heating in PW, leading to a swift increase in the carbon source’s temperature, often exceeding 2500–3000 °C within milliseconds. Plastics such as PE, PP, HDPE, and PET have been successfully demonstrated as suitable candidates for conversion into flash graphene. These thermoplastics, commonly found in consumer and industrial waste streams, are ideal candidates for FJH due to their compatibility with rapid heating and high carbon content.

Moreover, FJH is also effective in processing PVC, LDPE, and PS, expanding its applicability to a wide range of thermoplastics. Notably, FJH showed remarkable efficiency in handling mixed PW, reducing the need for meticulous sorting, and allowing large-scale upcycling of various feedstocks into valuable graphene products. The yields of flash graphene correlate with the thermal stability of the plastics. Polymers with higher thermal resistance tend to generate higher graphene yields while producing fewer volatile byproducts. Thermoplastics with higher thermal stability, such as HDPE and PET, tend to yield higher quantities of graphene due to their carbon structure and resistance to volatilization during the FJH process. Factors such as particle size, resistivity, and the addition of conductive agents like carbon black (CB) play critical roles in optimizing the yield and quality of flash graphene.

For instance, plastic powder with particle sizes ranging from 1 to 2 mm, is mixed with conductive additives such as carbon black (5 wt%). This combination ensures sufficient conductivity during the process, improving graphene yield. Conversely, larger particles may not achieve sufficient conductivity, while smaller particles may escape during the reaction, reducing efficiency. This scalability and adaptability make FJH a promising technique for addressing PW challenges while producing valuable nanomaterial [20,160]. The conversion of carbon-based starting materials into large quantities of turbostratic graphene through the FJH process has been successfully achieved by discharging direct current through large capacitors. However, challenges persist in extending this technique to other carbon allotropes, such as nanodiamonds and concentric carbon structures, and in achieving covalent functionalization of these allotropes. A notable breakthrough in this field is the production of fluorinated carbon materials through the FJH process. By facilitating the covalent attachment of fluorine to carbon species, this method has enabled the creation of fluorinated nanodiamonds (FNDs), turbostratic fluorinated graphene (TFG), and fluorinated concentric carbon (FCC) materials. These fluorinated phases exhibit unique electronic and structural properties, which make them highly valuable for applications such as advanced coatings, energy storage systems, and catalysis. Additionally, the FJH process offers precise control over the degree of fluorination, allowing for the tailoring of material properties to meet specific functional requirements [160]. Fluorinated carbon materials are characterized by their enhanced chemical stability, hydrophobicity, and resistance to high temperatures, making them ideal for demanding applications. For instance, in energy storage systems, fluorinated carbons like CFx are widely used as cathodes in lithium-ion and lithium primary batteries due to their high energy density, low self-discharge rates, and chemical resilience. They also find applications in supercapacitors, improving electrode wettability with specific electrolytes. Beyond energy storage, their hydrophobicity makes them excellent for protective coatings with anti-fouling and anti-corrosion properties, particularly in marine and industrial environments. Furthermore, their robustness and stability under harsh conditions make them valuable as catalyst supports in reactions such as fuel cell operations and chemical synthesis. In membrane technologies, fluorinated carbons are utilized in gas separation and proton exchange membranes, where resistance to aggressive chemicals is necessary. Moreover, low surface energy is advantageous for creating biomedical implants and biosensors where reduced protein adsorption is desirable [161,162].

In contrast, non-fluorinated carbon materials like graphene, CNTs, and activated carbon offer distinct advantages due to their high conductivity, mechanical strength, and versatility. These properties make them essential for applications in energy storage, structural materials, water purification, biomedical applications, and thermal management. For example, their high conductivity and large surface area are pivotal in enhancing the performance of supercapacitors and batteries. In structural applications, their lightweight yet robust nature makes them indispensable in the aerospace and automotive industries. Additionally, their high surface area, tunable porosity, and ease of functionalization make them suitable for water purification. Additionally, biomedical applications such as drug delivery, biosensing, and tissue engineering often rely on non-fluorinated carbons due to their ease of functionalization and better biocompatibility. Finally, thermal management applications, including the use of graphene and graphite in heat sinks and thermal interface materials, leverage their excellent conductivity and ability to dissipate heat efficiently [163,164,165]. In a previous study, researchers employed a solvent-free FJH approach to synthesize three distinct fluorinated carbon allotropes: fluorinated nanodiamonds (FND), fluorinated flash graphene (FFG), and fluorinated concentric carbon (FCC), as illustrated in Figure 12 [160]. This process was achieved by rapidly heating organic fluorine compounds and fluoride precursors for a short time.

The spectroscopic analysis confirmed alterations in the electronic states and identified a variety of short- and long-range structural configurations within the different fluorinated carbon allotropes. The study demonstrated a link between the heating duration and the resulting phases and compositions, showing that extended heating times led to the transformation of FND into FCC. This research demonstrates FJH as a fast, efficient, and cost-effective technique for producing fluorinated carbon allotropes in less than a second, while also allowing for the fine-tuning of their properties and structures to suit various applications.

Figure 15A illustrates the FJH setup, where various reactants are tightly packed inside a polytetrafluoroethylene (PTFE) tube, with tungsten carbide rods and graphite spacers serving as electrodes. The reactants consist of mixtures of different organic fluorine compounds. Using PTFE with 20 wt% carbon black (CB) as the reactant mixture allows the observation of distinct carbon material phases at different flash stages.

The proposed mechanism for forming these different carbon allotropes is presented in Figure 15B. During the FJH process, a relay-controlled discharge rapidly heats the reactant mixture to approximately 3000 K within milliseconds, followed by ultrafast cooling at an average rate of around 104 K/s. Throughout the flash duration, the fluoropolymer undergoes rapid heating, leading to the formation of FND particles, while the graphitization of FAC produces FFG. Extending the flash duration further induces surface graphitization of the FND particles, transforming them into FCC with a well-graphitized structure and polyhedral morphology, as illustrated in Figure 15C.

During the discharge, C–F bonds in the fluoropolymer break, leading to the release of fluorine-containing gases, such as HF. This process promotes the formation of C–C bonds indicated by Figure 15C and initiates the development of a carbon skeleton, followed by the growth of FND particles. The fluorination of carbon species plays a crucial role in forming FNDs, as the large electronegativity difference between carbon and fluorine (2.43) results in highly polarized covalent C–F bonds. This polarization decreases the electron density around the carbon atoms, facilitating the formation of sp^3^-hybridized carbons and additional C–C bonds, which stabilize the system by lowering its energy.

Another study for converting PW into FG through the application of FJH is shown in Figure 16 [20]. The process yielded FG, carbon oligomers, hydrogen, and light hydrocarbons. To produce high-quality graphene, a combination of alternating current (AC) and direct current (DC) flashes was used. First, the sample underwent AC-FJH treatment with an AC voltage of 120 V and a frequency of 60 Hz for approximately 8 s in a vacuum desiccator to facilitate outgassing as shown in Figure 16A. Following this, DC-FJH was performed using a capacitor bank consisting of 10 capacitors, each with a capacity of 450 V and 60 mF. These capacitors were charged to 110 V and discharged over 500 milliseconds, resulting in the production of enhanced FG. This FJH technique has proven to be effective in treating PW mixtures without the need for a catalyst, making it a suitable method for managing PW in landfills. The energy consumption was approximately 23 kJ/g, translating to an estimated cost of USD 125 per ton of PW, suggesting that the process is economically scalable.

Plastic powders with various particle sizes were tested to determine the optimal grain size for producing ACFG, with the best yield achieved using powders between 1 and 2 mm when mixed with 5 wt% carbon black (CB). Powders larger than 2 mm lacked sufficient conductivity to react effectively, while those smaller than 50 μm tended to escape from the quartz tube during the FJH process due to loose-fitting electrodes, significantly reducing the yield of AC-FG. As shown in Figure 16B, the AC-FG yield from HDPE powders of different sizes—2 mm, 1 mm, and 40 μm—was 23%, 21%, and 10%, respectively, when subjected to AC-FJH.

Another critical factor influencing the AC-FG yield is the initial resistivity across the sample. Compressing the plastic powder inside the quartz tube lowers the initial resistivity, thereby increasing the FG yield. Figure 16C illustrates the effect of varying initial resistivity, derived from different sample compressions between the electrodes, on the AC-FG yield from HDPE powder. AC-FJH was proven to be effective for producing FG from a range of thermoplastics, including PET, HDPE, PVC, LDPE, PP, and PS, with yields varying depending on the parent material’s properties. As indicated in Figure 13D, the yield of AC-FG correlated with the thermal stability of the original plastic, with more stable polymers yielding higher amounts of FG and generating fewer volatile oligomers. The yields were calculated based on the carbon content of the polymers, and the contribution of CB to FG conversion accounted for less than 4% of the total yield. Figure 16E displays large shreds of post-consumer HDPE plastic obtained from a commercial recycler, which were cut, mixed with 5 wt% CB, and subjected to FJH to produce AC-FG.

The yield of FG is significantly affected by the properties of the starting materials. AC-FJH was proven effective in generating FG from different thermoplastics, including PET, HDPE, PVC, LDPE, PP, and PS. The output of AC-FG is correlated with the thermal stability of the precursor materials; specifically, plastics with greater thermal resistance tend to produce higher yields of FG while generating fewer volatile oligomers. This relationship suggests that it is possible to convert PW into higher-value materials using minimal energy [166]. Adopting this strategy could lead to significant progress toward achieving plastic neutrality and reducing greenhouse gas emissions throughout the entire lifecycle, from initial production to the subsequent upcycling processes.

## 5. The Industrialization Progress of PW to Carbon-Based Nanomaterials

The industrialization of converting PW into carbon-based nanomaterials like graphene, CNTs, and porous carbon is advancing with promising technologies such as FJH, catalytic CVD, and pyrolysis. These methods can efficiently transform PW into high-value materials with applications in energy storage, water purification, and construction. However, challenges such as economic feasibility, feedstock heterogeneity, and regulatory concerns hinder widespread commercialization. Pilot-scale projects and government investments are fostering progress, with ongoing research focusing on cost reduction and scalability. Such efforts align with sustainability goals and a circular economy model, offering a pathway to mitigate plastic pollution. Key institutions like Rice University, particularly the research group of Professor James Tour, and companies like Universal Matter are at the forefront of this emerging field, where their focus is not just on making use of PW, but on any carbon-based waste material, converting it by FJH to FG [167] that can be applied in construction materials, cars, and other applications. In a recent study by Professor James Tour’s group [20], the cost of electricity required to convert one ton of PW to 180 kg of FG along with the volatile components was about USD 124 with no need for any catalyst, which is substantially better than recycling PW that is usually more expensive than producing new plastic. These promising results open the door for more scientific and technological advancements in how to deal with various waste materials including PW and overcome the challenges.

## 6. Conclusions

Plastic’s widespread use in modern society is due to its versatility and affordability, but the uncontrolled accumulation of PW presents a significant environmental and health challenge. Converting PW into high-value carbon materials offers a sustainable solution to this pressing issue. This review examines the current landscape of PW upcycling, emphasizing four main methods: thermal decomposition, FJH, CVD, and stepwise conversion.

Thermal decomposition, especially pyrolysis, is the most established and cost-effective method for producing carbon nanomaterials from PW, yielding high outputs at low costs that make pyrolysis economically viable for large-scale applications. FJH is noted for its rapid production of high-quality FG, with costs starting from USD 1 per kg, positioning it as a more cost-effective alternative to CVD while delivering materials with superior purity and fewer defects [20]. Stepwise conversion, although less common, involves a multi-stage process that is generally more expensive than pyrolysis but cheaper than CVD.

Despite these advancements, the current infrastructure for PW management is lacking, leading to inefficient waste processing. To improve economic feasibility, a significant shift in plastic upcycling technologies and the creation of value-added products are necessary. Additionally, strong policies aimed at reducing plastic production, encouraging reuse, and enhancing recycling efforts are essential.

Achieving plastic neutrality and promoting environmental sustainability requires coordinated efforts across scientific research, engineering, and policymaking. A collaborative, interdisciplinary approach is vital to addressing the complexities of PW management and advancing toward a circular plastic economy. The urgency of the PW crisis focuses on the need for innovative solutions and partnerships to tackle this global challenge effectively.

## Figures and Tables

**Figure 1 polymers-17-00063-f001:**
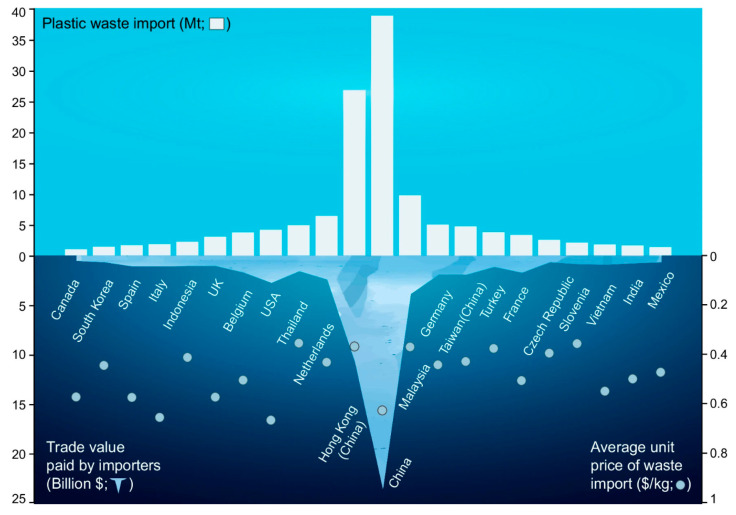
Total PW imports, total trade values, and average unit prices paid by the top 22 importers from 2013 to 2022. Reproduced from [7] under creative commons license CC BY 4.0.

**Figure 2 polymers-17-00063-f002:**
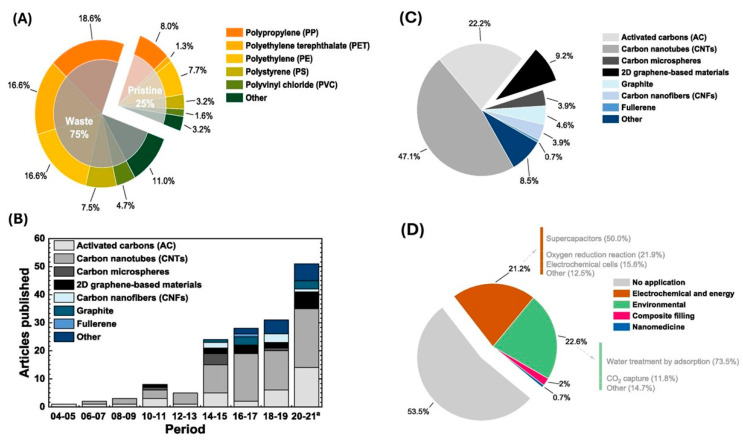
(**A**) Overview of the types and conditions of plastics utilized as feedstock for synthesizing carbon materials, including specifics on polymer types, physical state (e.g., unused, recycled), and any preprocessing methods applied. (**B**) Trend analysis shows the growth in the number of original research articles indexed in Scopus that focus on converting plastics into carbon materials over time, highlighting key milestones and shifts in research focus. a refers to the data collected by the authors of this publication on 29 June 2021 (**C**) Classification of the various carbon materials produced from plastics, detailing categories such as CNTs, graphene, and carbon black, along with their respective properties and characteristics. (**D**) Summary of the diverse fields of application for the carbon materials derived from plastics, including energy storage, electronics, environmental remediation, and composites, with examples of specific use cases in each sector. (**A**–**D**) are reproduced from [62] with permission from Elsevier, Copyright (2024).

**Figure 3 polymers-17-00063-f003:**
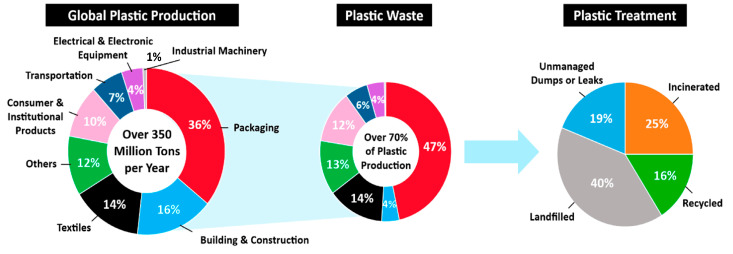
A comprehensive illustration depicting the lifecycle of PW, including its systematic fate from disposal to treatment practices. This includes an overview of different waste management techniques such as recycling, incineration, and landfilling, along with their environmental impacts and effectiveness in mitigating plastic pollution. Adapted from [69] under creative commons license CC BY 4.0.

**Figure 4 polymers-17-00063-f004:**
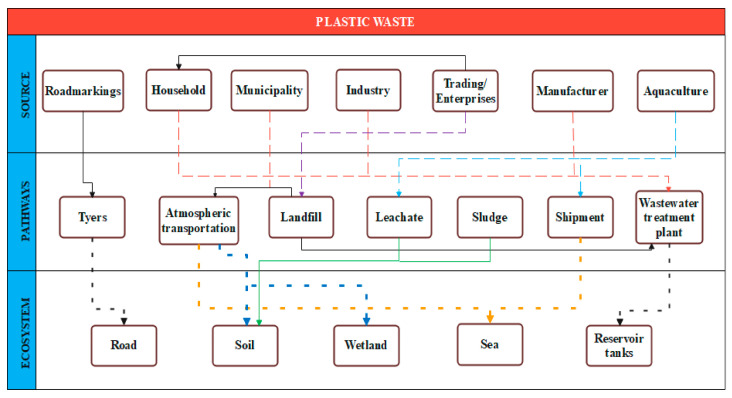
Sources and pathways of PW discharge, showing how waste from various origins (e.g., households, industry, aquaculture) reaches different environmental compartments (road, soil, wetlands, sea, reservoir tanks) via routes like atmospheric transport, landfills, sludge, and wastewater treatment. These lines refer to the different pathways of plastic discharge from various uses of plastic. Adapted from [70] under creative commons license CC BY 4.0.

**Figure 5 polymers-17-00063-f005:**
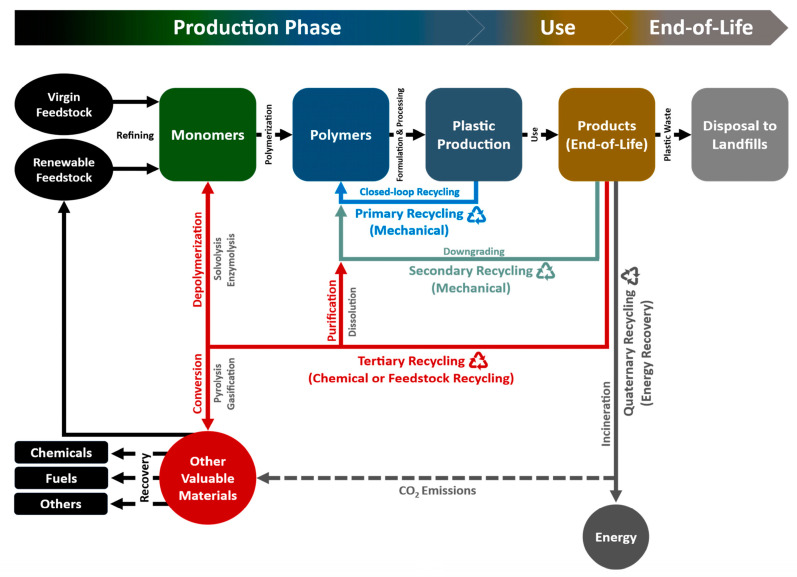
Possible routes for PW recycling, illustrating the four main pathways: Primary recycling, which involves mechanical reprocessing of scrap plastics into products with equivalent properties; secondary recycling, where used plastics are reprocessed into products with lower properties; tertiary recycling, focusing on the recovery of valuable chemical constituents such as monomers or additives; and quaternary recycling, which involves the recovery of energy from PW. These routes represent the key strategies in waste management and valorization of plastic materials. Adapted from [69] under creative commons license CC BY 4.0.

**Figure 6 polymers-17-00063-f006:**
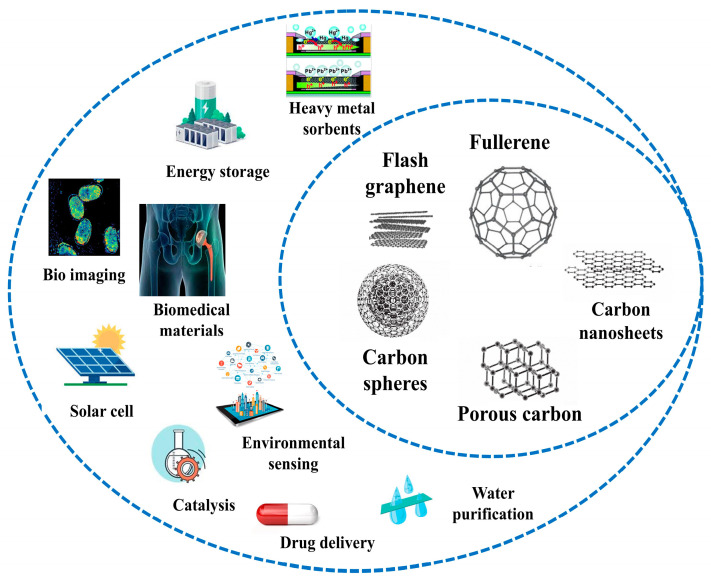
Diverse applications of carbon-based nanomaterials derived from the conversion of PW.

**Figure 8 polymers-17-00063-f008:**
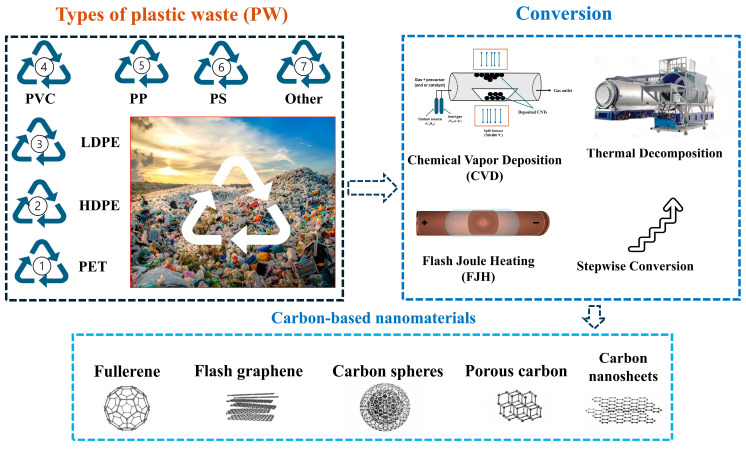
Schematic representation of the primary techniques employed for converting various types of PW into valuable graphene. This figure includes the following methods: (1) Thermal decomposition, illustrating the pyrolysis process and its parameters; (2) CVD, showing the setup and conditions necessary for graphene synthesis; (3) FJH, depicting the rapid heating mechanism and its advantages; and (4) stepwise conversion, outlining the multi-stage process involved in this method. Each technique draws attention to key features, potential benefits, and challenges associated with the conversion process.

**Figure 9 polymers-17-00063-f009:**
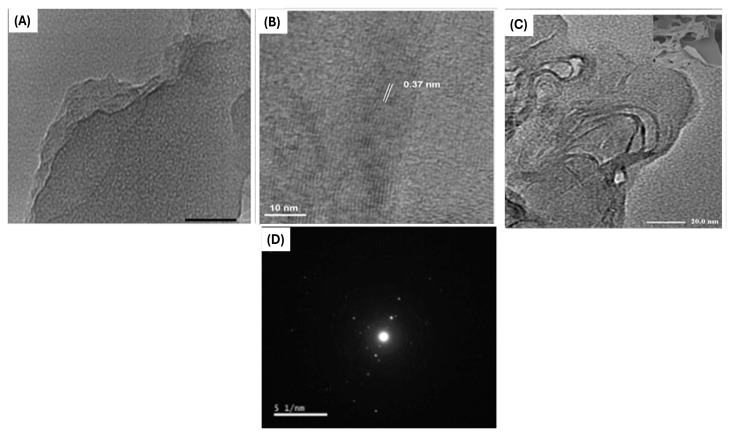
(**A**) Transmission electron microscopy (TEM) image and (**B**) high-resolution TEM (HR-TEM) image of graphene; (**C**) corresponding TEM image with an inset scanning electron microscopy (SEM) image of NG, illustrating its morphology; and (**D**) selected area electron diffraction (SAED) pattern of NG, providing insights into its crystallinity and structural integrity. Adapted from [148] under creative commons license CC BY 4.0.

**Figure 10 polymers-17-00063-f010:**
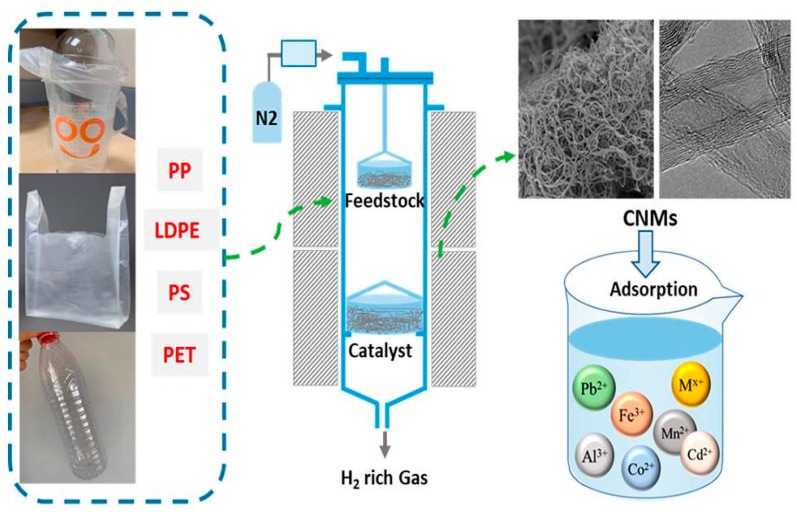
Overview of converting plastic packaging waste into carbon nanomaterials (CNMs) utilizing various catalyst materials. Reproduced from [149] with permission from American Chemical Society, Copyright (2022).

**Figure 13 polymers-17-00063-f013:**
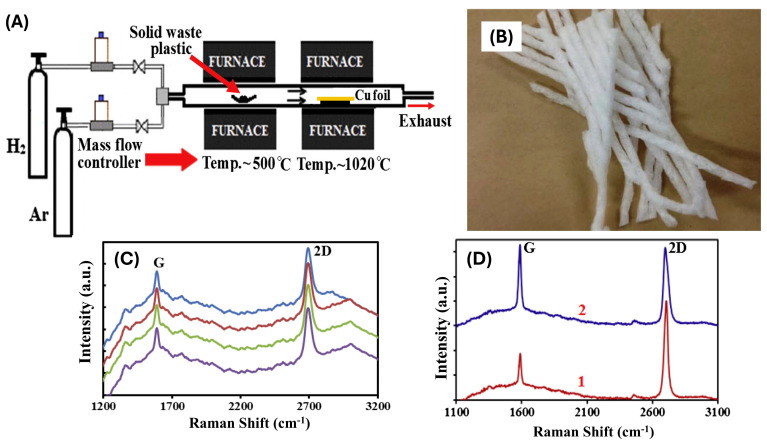
Synthesis of high-quality graphene from solid waste plastic using CVD. (**A**) Schematic representation of the CVD process, illustrating the key components and steps involved in the synthesis. (**B**) PW is utilized as the carbon feedstock, highlighting its role in the conversion process. (**C**) Raman spectra confirm the quality and characteristics of the graphene crystal. Colors represents the Raman spectra of the round graphene crystal, taken randomly at different points (places) on the crystal. (**D**) Raman spectra of the few-layer graphene crystals, providing insights into their structural integrity and layer count. (**A**–**D**) are reproduced from [157] with permission from Elsevier, Copyright (2014).

**Figure 14 polymers-17-00063-f014:**
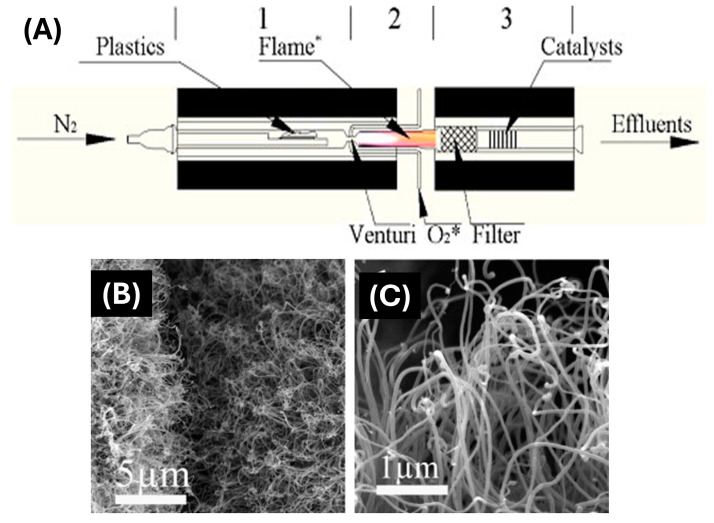
Synthesis of CNTs through sequential pyrolysis and combustion of PE. (**A**) Schematic representation of the three-stage stepwise CNT synthesis system, including Zone 1: pyrolysis, Zone 2: combustion, and Zone 3: synthesis stage. (*) the asterisk represents the flame that was present within the combustion stage only when oxygen was introduced to the venturi. (**B**,**C**) Scanning electron microscopy (SEM) images of CNTs generated from the combustion of HDPE at two different magnifications: 5 µm, illustrating overall morphology, and 1 µm, providing a closer view of the structural characteristics. (**A**–**C**) are reproduced from [158] with permission from Elsevier, Copyright (2010).

**Figure 15 polymers-17-00063-f015:**
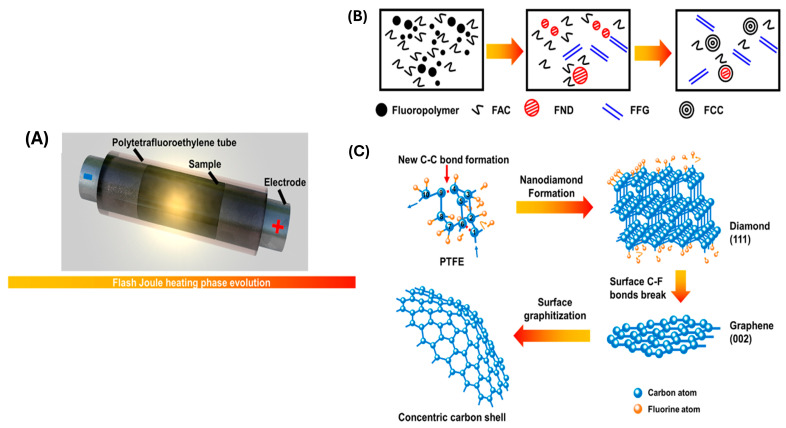
Ultrafast and controllable phase evolution through FJH. (**A**) Schematic representation of the FJH setup, illustrating the key components involved in the process. (**B**) Schematic depiction of the flash products at various stages of the FJH process, highlighting the impact of energy input on phase transitions. (**C**) Schematic illustration of the formation of FND from polytetrafluoroethylene (PTFE), followed by the subsequent conversion of FND into polyhedral FCC, showcasing the transformation sequence. Reprinted from [160] with permission from American Chemical Society, Copyright (2021).

**Figure 16 polymers-17-00063-f016:**
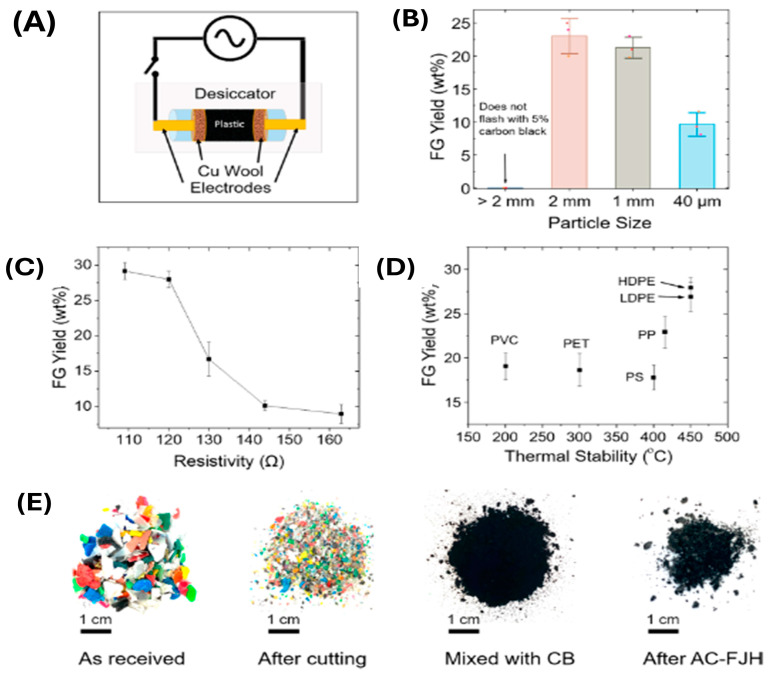
Conversion of PW into FG. (**A**) Schematic representation of the alternating current flash graphene (AC-FG) synthesis process, illustrating the key steps involved. (**B**) Graph showing the yield of AC-FG derived from HDPE. (**C**) Analysis of the impact of the initial resistivity of the HDPE/carbon black (CB) blend on the yield of AC-FG, highlighting the relationship between resistivity and output. (**D**) Typical AC-FG yields are obtained from various plastic types when the initial resistance is set to 120 Ω, showcasing the versatility of the method. (**E**) Sequential images depicting the transformation process from post-consumer plastic received from a recycler to the final conversion into FG using AC-FJH. Reprinted from [20] with permission from the American Chemical Society, Copyright (2020).

**Table 1 polymers-17-00063-t001:** Summary of techniques used for the conversion of PW into carbon-based materials.

Conversion Technique	Operation Condition			Produced Carbon Structure	Percent Carbon Yield (wt%)	Application	Ref.
	Heating Rate, (°C. min^−1^)	t, min	T, °C				
Discharge Arc method	N/A	20	2600	Nano-channeled ultrafine carbon tubes (NCUFCTs)/Multiwalled CNTs (MWCNTs)	N/A	Storage of energy	[123]
Catalysis pyrolysis	40	60	800		41.5	Catalysis	[124]
Thermal dissociation	8	60	800	Graphene	N/A	Adsorption	[102]
Catalytic carbonization	5	120	700	Porous carbon nanosheet	N/A	Supercapacitor	[125]
Autoclave-mediated pyrolysis	8	90	800	Graphene	N/A	Adsorption	[126]
Thermal-Hydrothermal method	N/A	1080	200	Carbon powder nanomaterials	72.3	Supercapacitor	[127]
One-pot microwave technique	N/A	5	N/A	Nitrogen-doped CNTs	N/A	Catalyst	[128]
Pyrolysis	50	N/A	850	Porous carbon	20–25	Adsorption	[129]
Chemical treatment	N/A	60	N/A	CNTs aerogels	N/A	Sorption	[130]
Catalytic CVD	15	N/A	550	Multiwalled CNTs	N/A	Supercapacitor	[131]
Catalytic carbonization	N/A	N/A	N/A	Porous carbon nanosheet	N/A	Supercapacitor	[132]
Carbonization	N/A	60	1000	Porous carbon	15	CF4 Adsorption	[133]
Chemical depolymerization	N/A	360	1000	N-doped mesoporous carbon (NPC)	N/A	Supercapacitor	[134]
Carbonization and stepwise cross-linking	10	600	280	N-doped mesoporous carbon (NPC)	N/A	CO_2_ adsorption	[135]
Pressurized carbonization	20	120	700	Porous carbon sheet	33.4	N/A	[136]
Low-temperature carbonization and etching	5	60	500	Hierarchical porous carbon	N/A	Supercapacitor	[137]
Carbonization and chemical activation	N/A	N/A	700	Nanoporous carbon	N/A	Adsorption	[138]

Notes: N/A—not available.

**Table 2 polymers-17-00063-t002:** Summary of carbonization processes for synthesizing carbon materials from plastic.

Carbonization Process	Conditions		Polymer Type	Produced Carbon Structure	Ref.
	Carbonization Temp, °C	Atmosphere			
Anoxic pyrolysis carbonization (no stabilization)	500–1000	Inert	PET, phenol formaldehyde resin (PFR)	Amorphous carbon (Activated carbon, mesoporous carbon, and carbon fibers)	[146]
Anoxic pyrolysis carbonization (oxidated stabilization)	500–1000	Inert	PAN, LDPE, PVC		
	Oxidation temperature 200–300 °C				
Anoxic pyrolysis carbonization (chemical stabilization)	Friedel–Crafts reaction	N/A	PE, PS		
Catalytic carbonization	400–900	Inert, in the presence of a metal compound catalyst	PE, PP, PS, PFR, PVC, PVDF, PTFE, PVA, PET	Graphitic carbons (CNTs, CNFs, carbon nanosheets, graphene, carbon spheres, carbon foam	[146]
Pressure carbonization (Pressurized atmosphere)	600–850	Pressurized	PE, PP, PS, PVC	Amorphous carbons (activated carbon, carbon dots, carbon spheres)	[147]
Pressure carbonization (Hydrothermal carbonization )	150–300	N/A	PVC	Amorphous carbons (activated carbon, carbon dots, carbon spheres)	[147]

Notes: N/A—not available.

## Data Availability

Data will be made available upon request.

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
