# Peer review of "From Waste to Worth: Upcycling Plastic into High-Value Carbon-Based Nanomaterials"

_polymers, 2024, doi:10.3390/polym17010063_

Round 1
Reviewer 1 Report
Comments and Suggestions for Authors
This review presents the progress of the plastics upcycling, from waste to worth, which is a meaningful work. However, the whole manuscript was present in a good organizing but bad writing. The review does not recommend publication at this stage.
(1) English language needs extensive editing, such as “. exploring more innovative applications.”(line 52)
(2) Line 53-57, what is the difference between nanomaterials of (ii) and carbon-based materials with diverse nanostructures of (iii)? the classification is confused.
(3) Line 59, PET should defined for its first appeared.
(4) Line 64, how can PET exhibits excellent electrical and thermal properties?
(5) The main topic of this work is the upcycling of plastic, rather than PET, ONLY PET was discussed in the Introduction, which should revised.
(6) Line 153-154, the PVC, PP, and PS has been defined in line 98-99, thus the abbreviations should used in the following manuscript.
(7) A section about the conversion mechanism of PW to carbon-based nanomaterials should added, such as graphene, CNT, fullerene, spheres, and porous carbon.
(8) Graphene in the title of section 3 should revised to Carbon.
(9) The abbreviations of HDPE, LDPE, PF, PMMA should defined.
(10) Table 2, why does there is no produced carbon structure and reference for the first carbonization process?
(11) How is the industrialization progress of PW to carbon-based nanomaterials?

English language needs extensive editing
Author Response
RESPONSE TO THE REVIEWER COMMENTS:
Manuscript ID: polymers-3319020
Title: From waste to worth: upcycling plastic into high-value carbon-based nanomaterials.
Response to reviewers:
To editor:
Dear respected editor Cicilia Zhang
Thank you for your time and effort and reviewer comments. The reviewers made a careful and professional review of our manuscript: polymers-3319020. The suggestions are helpful for us to refine this review. We carefully read the reviewer's comments and suggestions and now completed a revision of the manuscript that addresses their concerns. Thorough and extensive revisions have been made throughout the manuscript following the suggestions from the reviewers. We thank you and the reviewers for the constructive suggestions that have improved both the quality and the clarity of the manuscript, and we believe that the revised review is now acceptable for publication in polymers.
Thank you, and we anticipate your prompt feedback.
Academic editor notes:
Reviewers has identified several deficiencies. Further, the writing needs to be strongly improved.
This manuscript will need some revision for accuracy and clarity. For example, the heading "Literature Survey" would better be "Previous Results." In the first sentence under that heading, "Recent publications have highlighted" should be omitted. The senrence should be, "Various innovative ... have been highlighted ..."
Author response: Thank you so much for your comment. The heading has been renamed and now it is “Previous results” on page 6. Also, the first sentence has been corrected as suggested on the same page (page 6).
“2. Previous results
Various innovative methods have been highlighted for transforming PW into valuable carbon-based materials, marking significant progress in the field.”
Reviewer 1
This review presents the progress of the plastics upcycling, from waste to worth, which is a meaningful work. However, the whole manuscript was present in a good organizing but bad writing. The review does not recommend publication at this stage.
(1) English language needs extensive editing, such as “. exploring more innovative applications.”(line 52)
Author response: Thank you so much for your comment. This sentence has been corrected on page 2, and the whole manuscript has been linguistically checked.
Inserted section
“Recent studies have moved beyond the traditional use of PW in bitumen [1], and food production [2] to more advanced applications.”
(2) Line 53-57, what is the difference between nanomaterials of (ii) and carbon-based materials with diverse nanostructures of (iii)? the classification is confused.
Author response: Thank you so much for your comment. The whole paragraph has been rewritten after removing the mentioned “nanomaterials” and focusing only on carbon-based materials on page 2.
Inserted section
“Upcycling reduces waste accumulation while simultaneously creating economic opportunities by transforming discarded materials into higher-value products, making it an effective solution to address the challenges posed by plastic waste. In this context, utilizing PW as a raw material for producing value-added products and materials has been proposed to enhance the economic viability of plastics recycling. Recent studies have moved beyond the traditional use of PW in bitumen [1], and food production [2] to more advanced applications. Such applications include the generation of gaseous and liquid fuels, as well as valuable chemicals [3, 4]. Additionally, upcycling of PW into carbon-based materials with diverse nanostructures and morphologies such as hierarchical porous carbon, carbon nanotubes (CNTs), graphene, carbon quantum dots, and carbon-based composites [5-14] is quite useful in different applications such as water treatment.”
(3) Line 59, PET should defined for its first appeared.
Author response: Thank you so much for your comment. The word PET, which is polyethylene terephthalate, has been defined on page 2.
Inserted section
“Polyethylene terephthalate (PET) can be upcycled to various bio-polymers and in this regard, Tiso et al. [15] converted PET waste to polyhydroxyalkanoate through thermal hydrolysis followed by degrading the produced terephthalate microbiologically using Pseudomonas umsongensis GO16.”
(4) Line 64, how can PET exhibits excellent electrical and thermal properties?
Author response: Thank you so much for your comment. This sentence has been removed from the manuscript because the whole paragraph has been rewritten in a different way to highlight the upcycling of plastic waste.
(5) The main topic of this work is the upcycling of plastic, rather than PET, ONLY PET was discussed in the Introduction, which should revised.
Author response: Thank you so much for your comment. The upcycling of PET and other types of plastic waste has been added to the introduction on page 2.
Inserted section
“Polyethylene terephthalate (PET) can be upcycled to various bio-polymers and in this regard, Tiso et al. [15] converted PET waste to polyhydroxyalkanoate through thermal hydrolysis followed by degrading the produced terephthalate microbiologically using Pseudomonas umsongensis GO16. Also, they used the same bacterial strain to produce bio-based poly(amide urethane), which was a new bioplastic. Polystyrene (PS) is another common plastic material that could be upcycled to bio-oils owing to its high yield of that sort of oils that can be up to 84wt% via pyrolysis [16]. Such a yield is substantially higher than that of other types of plastics such as polyethylene (PE) with about 43wt% of bio-oil yield [17]. Polyvinyl chloride (PVC) could be dechlorinated to produce high density polyethylene (HDPE) waxes through a heterogeneous catalytic reaction using a carbon-supported platinum catalyst [18]. These waxes are more valuable than PVC itself. Upcycling of polypropylene (PP) could be useful in water treatment as proved by Li et al. [19] who co-pyrolyzed pp with a dried biomass sample of Microcystis aeruginosa using potassium carbonate as an activator to produce a cyanobacteria-plastic carbon composite. This composite showed promising results in adsorbing methylene blue with an adsorption capacity of 490 mg/g owing to its high surface area (2135 m2/g). Co-pyrolysis has also been used to produce carbon materials such as biochar with high rate of bio-oil production by pyrolyzing biomass such as pinecone with PW such as low density polyethylene (LDPE), PP, and PS at 500 °C [20]. Through a three-step process comprises air-bromination, dehydrobromination, and olefin metathesis reactions, Zeng et al. [21] upcycled PE to α,ω-divinyl-functionalized oligomers, which can be utilized in producing chemical additives, lubricants, waxes, and plastics as well.”
(6) Line 153-154, the PVC, PP, and PS has been defined in line 98-99, thus the abbreviations should used in the following manuscript.
Author response: Thank you so much for your comment. The abbreviations have been corrected and continuously used throughout the whole manuscript.
(7) A section about the conversion mechanism of PW to carbon-based nanomaterials should added, such as graphene, CNT, fullerene, spheres, and porous carbon.
Author response: Thank you so much for your comment. The conversion of plastic waste to different types of carbon materials is classified based on the treatment (conversion) technique (eg: flash joule heating). The explanation of the conversion processes or mechanisms was already provided in each treatment or conversion method followed by detailed examples from the literature. In this regard, Figure 13.a explains the synthesis of high-quality graphene from solid waste plastic using chemical vapor deposition, Figure 14.a explains the synthesis of carbon nanotubes (CNTs) through sequential pyrolysis and combustion of polyethylene, Figure 15.c illustrates of the formation of fluorinated nanodiamonds from polytetrafluoroethylene, followed by the subsequent conversion of these nanodiamonds into polyhedral fluorinated concentric carbon, showcasing the transformation sequence, and Figure 16.a shows the conversion of plastic waste into flash graphene using alternating current (AC).
(8) Graphene in the title of section 3 should revised to Carbon.
Author response: Thank you so much for your comment. The word “graphene” has been changed to “carbon” on page 9.
“Carbon-based materials produced from PW conversion.”
(9) The abbreviations of HDPE, LDPE, PF, PMMA should defined.
Author response: Thank you so much for your comment. High density polyethylene (HDPE), low density polyethylene (LDPE), phenol formaldehyde resin (PFR), and poly(methyl methacrylate) (PMMA) have all been defined in the manuscript on pages 2, 12, and 15.
(10) Table 2, why does there is no produced carbon structure and reference for the first carbonization process?
Author response: Thank you so much for your comment. The produced carbon structure and the reference of the first carbonization process are the same of the second and the third processes in Table 2. This has been clarified by adding all the borders to the table, so it is now much clearer than it was before.
Modified table
|
Carbonization process |
Conditions |
Polymer type |
Produced carbon structure |
Ref. |
|
|
Carbonization Temp, ℃ |
Atmosphere |
||||
|
Anoxic pyrolysis carbonization (no stabilization) |
500-1000 |
Inert |
PET, phenol formaldehyde resin (PFR) |
Amorphous carbon (Activated carbon, mesoporous carbon, and carbon fibers) |
[22] |
|
Anoxic pyrolysis carbonization (oxidated stabilization) |
500-1000 |
Inert |
PAN, LDPE, PVC |
||
|
Oxidation temperature 200-300 ℃ |
|||||
|
Anoxic pyrolysis carbonization (chemical stabilization) |
Friedel–Crafts reaction |
N/A |
PE, PS |
||
|
Catalytic carbonization |
400-900 |
Inert, in the presence of metal compound catalyst |
PE, PP, PS, PFR, PVC, PVDF, PTFE, PVA, PET |
Graphitic carbons (carbon nanotubes, carbon nanofibers, carbon nanosheets, graphene, carbon spheres, carbon foam |
[22] |
|
Pressure carbonization (Pressurized atmosphere) |
600-850 |
Pressurized |
PE, PP, PS, PVC |
Amorphous carbons (activated carbon, carbon dots, carbon spheres) |
[23] |
|
Pressure carbonization (Hydrothermal carbonization ) |
150-300 |
N/A |
PVC |
Amorphous carbons (activated carbon, carbon dots, carbon spheres) |
[23] |
(11) How is the industrialization progress of PW to carbon-based nanomaterials?
Author response: Thank you so much for your comment. A new section about the industrialization progress with an example showing promising scientific and technological advancements has been added on page 24.
Inserted text
“4. The industrialization progress of PW to carbon-based nanomaterials
The industrialization of converting PW into carbon-based nanomaterials like graphene, CNTs, and porous carbon is advancing with promising technologies such as FJH, catalytic CVD, and pyrolysis. These methods can efficiently transform plastic waste into high-value materials with applications in energy storage, water purification, and construction. However, challenges such as economic feasibility, feedstock heterogeneity, and regulatory concerns hinder widespread commercialization. Pilot-scale projects and government investments are fostering progress, with ongoing research focusing on cost reduction and scalability. Such efforts align with sustainability goals and a circular economy model, offering a pathway to mitigate plastic pollution. Key institutions like Rice University, particularly the research group of Professor James Tour and companies like Universal Matter are at the forefront of this emerging field, where their focus is not just on making use of PW, but on any carbon-based waste material to convert it by FJH to FG [24] that can be applied in construction materials, cars, and other applications. In a recent study by Professor James Tour group [25], the cost of electricity required to convert one ton of PW to 180 kg of FG along with the volatile components was about 124 US dollars with no need for any catalyst, which is substantially better than recycling PW that is usually more expensive than producing new plastic. These promising results open the door for more scientific and technological advancements in how to deal with various waste materials including PW and overcome the challenges.

Reviewer 2 Report
Comments and Suggestions for Authors
Abstract:
-
"Plastic waste (PW) has become a significant environmental challenge due to its persistent accumulation and detrimental impact on ecosystems."
- Comment: This line is informative but could benefit from quantitative data to emphasize the severity of PW accumulation.
-
"This study explores innovative approaches to convert PW into high-value carbon nanomaterials (CNMs) such as graphene, carbon nanotubes (CNTs), and other advanced carbon structures."
- Comment: Specific methodologies should be briefly listed here to give readers a quick overview of the techniques employed.
-
"Four primary methodologies are discussed: thermal decomposition, chemical vapor deposition (CVD), flash joule heating (FJH), and stepwise conversion."
- Comment: It would be helpful to briefly mention how these methods differ in their efficiency or application potential for better clarity.
-
"This research highlights the growing potential of upcycling PW into valuable materials that align with global sustainability goals."
- Comment: Consider defining specific goals from the United Nations’ Sustainable Development Goals (SDGs) to link to global sustainability more concretely.
Keywords:
- Keywords are well-chosen and relevant. However, consider adding "Upcycling," "Sustainability," or "Green Technology" for broader indexing.
Introduction:
-
"The global production of plastics has seen a steep rise over the years. In 2019, nearly 370 million metric tons of plastic were produced worldwide..."
- Comment: Good start with statistics; however, specify if this includes all types of plastics or if it’s focused on single-use plastics for clarity.
-
"Projections indicate that by 2050, over 25,000 million metric tons of PW will have been generated."
- Comment: This forecast is impactful but could be further contextualized by mentioning the current or expected recycling rates to highlight the scale of the issue.
-
"These predictions highlight the ongoing issue of mismanagement, largely due to the limitations of current recycling and reuse technologies."
- Comment: Elaborate briefly on specific limitations to provide a stronger foundation for discussing upcycling technologies later.
-
"Each year, 284 million metric tons of PW are generated, with over 2.41 million tons ending up in the ocean."
- Comment: Consider breaking down the sources of oceanic plastic waste, as it provides depth to the argument on the need for recycling infrastructure improvement.
-
"Through chemical and physical processes, this waste breaks down into microplastics smaller than 5 mm, releasing over 10 million tons of these pollutants into the environment annually."
- Comment: Excellent point. Adding information about the environmental or health risks of microplastics could emphasize the urgency.
Methodology:
Thermal Decomposition:
-
"Thermal decomposition is a chemical process that occurs when a substance is exposed to elevated temperatures, causing it to break down into simpler components."
- Comment: The definition is clear. However, mention specific temperature ranges relevant to this study for contextual clarity.
-
"This process yields valuable chemicals in the form of pyrolytic gases and oils, alongside heat energy."
- Comment: Consider detailing the types of chemicals produced and their potential applications. This could enhance the relevance of thermal decomposition as a methodology.
Chemical Vapor Deposition (CVD):
-
"CVD is a sophisticated technique widely used to produce high-quality solid materials, particularly in controlled vacuum environments."
- Comment: This description is well-executed. Including a diagram or schematic of the CVD process could enhance understanding for readers unfamiliar with the technique.
-
"The recycling of PW into valuable materials was achieved using a cost-effective, simple, and dependable CVD system."
- Comment: The terms "cost-effective" and "simple" should be quantified or qualified, if possible. Adding data or specific cost comparisons strengthens the claim.
Flash Joule Heating (FJH):
-
"FJH is an advanced technique for producing high-quality carbon materials, particularly for generating top-tier carbon nanomaterials."
- Comment: This is clear, but defining "top-tier" could avoid ambiguity. Mention specific properties (e.g., purity, structure) that make FJH-produced materials superior.
-
"This cutting-edge method leverages electrical energy to trigger rapid joule heating in PW, leading to a swift increase in the carbon source's temperature."
- Comment: Adding a temperature range achieved in this process would be beneficial to understand the energy dynamics involved.
Stepwise Conversion:
-
"The stepwise conversion method is an adaptable and efficient process designed to transform PW into valuable carbon materials through a multi-stage approach."
- Comment: This introduction is clear, but specify the unique benefits or challenges compared to single-step methods to provide context for its “efficiency.”
-
"This method not only provides a sustainable approach to managing PW but also upcycles waste into high-value products, demonstrating its potential in both environmental remediation and advanced material production."
- Comment: This concluding line could emphasize the scalability of stepwise conversion, especially if industrial application is intended.
Author Response
RESPONSE TO THE REVIEWER COMMENTS:
Manuscript ID: polymers-3319020
Title: From waste to worth: upcycling plastic into high-value carbon-based nanomaterials.
Response to reviewers:
To editor:
Dear respected editor Cicilia Zhang
Thank you for your time and effort and reviewer comments. The reviewers made a careful and professional review of our manuscript: polymers-3319020. The suggestions are helpful for us to refine this review. We carefully read the reviewer's comments and suggestions and now completed a revision of the manuscript that addresses their concerns. Thorough and extensive revisions have been made throughout the manuscript following the suggestions from the reviewers. We thank you and the reviewers for the constructive suggestions that have improved both the quality and the clarity of the manuscript, and we believe that the revised review is now acceptable for publication in polymers.
Thank you, and we anticipate your prompt feedback.
Academic editor notes:
Reviewers has identified several deficiencies. Further, the writing needs to be strongly improved.
This manuscript will need some revision for accuracy and clarity. For example, the heading "Literature Survey" would better be "Previous Results." In the first sentence under that heading, "Recent publications have highlighted" should be omitted. The senrence should be, "Various innovative ... have been highlighted ..."
Author response: Thank you so much for your comment. The heading has been renamed and now it is “Previous results” on page 6. Also, the first sentence has been corrected as suggested on the same page (page 6).
“2. Previous results
Various innovative methods have been highlighted for transforming PW into valuable carbon-based materials, marking significant progress in the field.”
Reviewer 2
Abstract:
- "Plastic waste (PW) has become a significant environmental challenge due to its persistent accumulation and detrimental impact on ecosystems."
- Comment:This line is informative but could benefit from quantitative data to emphasize the severity of PW accumulation.
Author response: Thank you so much for your comment. The amount of produced plastics in 2024 along with the expected amount of plastics waste by 2060 has been added to the abstract on page 1.
Inserted text
“According to the United Nations Environment Program (UNEP), the amount of plastics produced in 2024 is around 500 million tons, most of it is projected to end up as waste and result in billions of tons of PW by 2060 in the absence of effective measures that can solve this global issue.”
Abstract:
- "This study explores innovative approaches to convert PW into high-value carbon nanomaterials (CNMs) such as graphene, carbon nanotubes (CNTs), and other advanced carbon structures."
- Comment: should be briefly listed here to give readers a quick overview of the techniques specific methodologies employed.
Author response: Thank you for your comment. The specific techniques were added.
Inserted text
“This study explores innovative approaches to convert PW into high-value carbon nanomaterials (CNMs) such as graphene, carbon nanotubes (CNTs), and other advanced carbon structures. Various methods including pyrolysis, arc discharge, catalytic degradation, laser ablation have been investigated in transforming PW into CNMs.”
- "Four primary methodologies are discussed: thermal decomposition, chemical vapor deposition (CVD), flash joule heating (FJH), and stepwise conversion."
- Comment:It would be helpful to briefly mention how these methods differ in their efficiency or application potential for better clarity.
Author response: Thank you, we further elaborated on the differences between these methods.
Modified text
“However, four primary methodologies are discussed: thermal decomposition, chemical vapor deposition (CVD), flash joule heating (FJH), and stepwise conversion. The scalability of the discussed pathways for industrial applications varies significantly. Thermal decomposition, particularly pyrolysis, is highly scalable due to its straightforward setup and cost-effective operation, making it suitable for large-scale waste processing plants. It also produces fuel byproducts that can be used as an alternative source of energy promoting the concept of energy recovery and circular economy. CVD, while producing high-quality carbon materials, is less scalable due to the high cost and required complex equipment, catalyst, high temperature, and pressure which limits its use to specialized applications. FJH offers rapid synthesis of high-quality graphene by an economically viable technique that can also generate valuable products such as green hydrogen, carbon oligomers, and light hydrocarbons. However, it still requires optimization for industrial throughput. Stepwise conversion, involving multiple stages, can be challenging to scale due to higher operational complexity and cost, but it offers precise control over material properties for niche applications.”
- "This research highlights the growing potential of upcycling PW into valuable materials that align with global sustainability goals."
- Comment:Consider defining specific goals from the United Nations’ Sustainable Development Goals (SDGs) to link to global sustainability more concretely.
Author response: Thank you for this note, we added the specific SDGs that align with PW upcycling.
Inserted text
“This research highlights the growing potential of upcycling PW into valuable materials that align with global sustainability goals including industry, innovation, and infrastructure (Goal 9), sustainable cities and communities (Goal 11), and responsible consumption and production (Goal 12).”
Keywords:
- Keywords are well-chosen and relevant. However, consider adding "Upcycling," "Sustainability," or "Green Technology" for broader indexing.
Author response: Thank you for your comment, we added the word "Upcycling".
Introduction:
- "The global production of plastics has seen a steep rise over the years. In 2019, nearly 370 million metric tons of plastic were produced worldwide..."
- Comment: Good start with statistics; however, specify if this includes all types of plastics or if it’s focused on single-use plastics for clarity.
Author response: Thank you, we included the types of plastics involved in these statistics.
Inserted text
“The global production of plastics has seen a steep rise over the years. In 2019, nearly 370 million metric tons of plastic were produced worldwide including thermoplastics, thermosets, polyurethanes, adhesives, coatings and sealants, elastomers, and PP-fibers. (polyacrylic fibers, PET fibers, and PA fibers are not included).”
- "Projections indicate that by 2050, over 25,000 million metric tons of PW will have been generated."
- Comment: This forecast is impactful but could be further contextualized by mentioning the current or expected recycling rates to highlight the scale of the issue.
Author response: The current and expected recycling rates are now available in the paragraph.
Inserted text
The EU Directive 94/62/EC requires EU countries to recycle 50% of plastic by 2025 and 55% by 2030. However, countries couldn’t achieve this target, Germany, for example, a country that has a modern and well-established plastic waste recycling system in the form of the dual system only achieves recycling rates of less than 30% for post-consumer plastics. This highlights the limitations of mechanical recycling in meeting the ambitious EU goal for plastic recycling. Therefore, new recycling concepts and technologies are desperately needed to complement or substitute traditional recycling in order to increase the circularity of PW and achieve recycling targets [26].
- "These predictions highlight the ongoing issue of mismanagement, largely due to the limitations of current recycling and reuse technologies."
- Comment: Elaborate briefly on specific limitations to provide a stronger foundation for discussing upcycling technologies later.
Author response: The specific limitations of current recycling and reuse technologies were added.
Inserted text
These predictions highlight the ongoing issue of mismanagement, largely due to the limitations of current recycling and reuse technologies as they can only process certain types of plastic waste which leaves much PW discarded in landfills. Moreover, traditional recycling can only be done for limited times otherwise the produced plastic loses its quality. Recycled plastic often has lower quality than the new one and it requires pretreatment steps of cleaning, careful sorting, drying which is time-consuming and needs additional cost leading to inadequate and ineffective recycling rates [27].
- "Each year, 284 million metric tons of PW are generated, with over 2.41 million tons ending up in the ocean."
- Comment: Consider breaking down the sources of oceanic plastic waste, as it provides depth to the argument on the need for recycling infrastructure improvement.
Author response: The sources of oceanic plastic waste were added to the text.
The majority of oceanic plastic is from land sources as between 70% and 80% of its weight is carried by rivers or coastlines from land to the sea [28]. The remaining 20% to 30% originates from marine sources including lines, ropes, abandoned ships, and fishing nets [29].
- "Through chemical and physical processes, this waste breaks down into microplastics smaller than 5 mm, releasing over 10 million tons of these pollutants into the environment annually."
- Comment: Excellent point. Adding information about the environmental or health risks of microplastics could emphasize the urgency.
Author response: Thank you, we added information about the environmental and health risks of microplastics.
Inserted text
Microplastics can severely affect human health as it can enter the body through ingestion, skin contact and inhalation [30]. The potential harmful impacts come from the leaching of its inorganic and organic toxic chemical components and its ability to adsorb hazardous extraneous substances through the biological or chemical vectors associated with it. Moreover, it can cause direct physical damage from ingested particles of plastic debris that attach in various organs [31].
Thermal Decomposition:
- "Thermal decomposition is a chemical process that occurs when a substance is exposed to elevated temperatures, causing it to break down into simpler components."
- Comment: The definition is clear. However, mention specific temperature ranges relevant to this study for contextual clarity.
Author response: Thank you, we have mentioned the specific temperature ranges.
Inserted text
Thermal decomposition is a chemical process that occurs when a substance is exposed to elevated temperatures ranging from 400 to 600 .
- "This process yields valuable chemicals in the form of pyrolytic gases and oils, alongside heat energy."
- Comment: Consider detailing the types of chemicals produced and their potential applications. This could enhance the relevance of thermal decomposition as a methodology.
Author response: The types of the produced chemicals and their potential applications were added.
Inserted text
Plastic waste pyrolysis can produce hydrogen Fuel, diesel that can be used to run engines for industry and transport, and crude oil that can be refined into gasoline. Products from pyrolysis are frequently used as cogeneration oil, engine fuel, solid fuel, and raw materials for additional processing to make polymers and other goods [32].
Chemical Vapor Deposition (CVD):
- "CVD is a sophisticated technique widely used to produce high-quality solid materials, particularly in controlled vacuum environments."
- Comment: This description is well-executed. Including a diagram or schematic of the CVD process could enhance understanding for readers unfamiliar with the technique.
Author response: Thank you for your valuable comment; we added a schematic of the CVD process after that paragraph.
Inserted figure
Figure 11. Schematic representation of the chemical vapor deposition setup showing the main components and stages
- "The recycling of PW into valuable materials was achieved using a cost-effective, simple, and dependable CVD system."
- Comment: The terms "cost-effective" and "simple" should be quantified or qualified, if possible. Adding data or specific cost comparisons strengthens the claim.
Author response: Thank you for your comment. The terms were quantified by adding data from recent literature.
Inserted text
The recycling of PW into valuable materials was achieved using a cost-effective, simple, and dependable CVD system. A recent economic evaluation study compared the upcycling CVD process with the traditional method of producing carbon nanotubes. The cost of the upcycling process was 2,999 $ kg− 1 which was slightly higher than the conventional one of 2,930 $ kg− 1 but a 20% reduction of the argon flow rate made the upcycling technique surpass the conventional process [33].
Flash Joule Heating (FJH):
1."FJH is an advanced technique for producing high-quality carbon materials, particularly for generating top-tier carbon nanomaterials."
Comment: This is clear, but defining "top-tier" could avoid ambiguity. Mention specific properties (e.g., purity, structure) that make FJH-produced materials superior.
Author response: The authors deeply acknowledge the valuable comment given by the reviewer. The term “top-tier” has been clarified.
Modified word
FJH is an advanced technique for producing high-quality carbon materials, particularly for generating high-performance carbon nanomaterials.
2."This cutting-edge method leverages electrical energy to trigger rapid joule heating in PW, leading to a swift increase in the carbon source's temperature."
Comment: Adding a temperature range achieved in this process would be beneficial to understand the energy dynamics involved.
Author response: The authors deeply acknowledge the valuable comment given by the reviewer. The manuscript has been updated to include the temperature range for FJH process.
Inserted section
This cutting-edge method leverages electrical energy to trigger rapid joule heating in PW, leading to a swift increase in the carbon source's temperature, often exceeding 2,500–3,000°C within milliseconds.
Stepwise Conversion:
1."The stepwise conversion method is an adaptable and efficient process designed to transform PW into valuable carbon materials through a multi-stage approach."
Comment: This introduction is clear, but specify the unique benefits or challenges compared to single-step methods to provide context for its “efficiency.”
Author response: The authors deeply acknowledge the valuable suggestion given by the reviewer. The manuscript has been revised to highlight the benefits and challenges of stepwise conversion methods compared to single-step approaches, with an emphasis on process efficiency.
Inserted section
Compared to single-step methods such as thermal decomposition or flash Joule heating, the stepwise approach offers unique advantages and presents specific challenges. Several benefits distinguish the stepwise method from its single-step counterparts. First, the separation of decomposition and synthesis stages allows for fine-tuning of reaction conditions, resulting in carbon materials with specific pore structures or dimensions, which is particularly advantageous for CNT production. Additionally, the use of targeted catalysts ensures efficient utilization of hydrocarbons, leading to higher yields and better material quality. The method also incorporates energy recovery from exothermic reactions, enhancing overall sustainability and reducing energy consumption. Despite its advantages, the stepwise conversion method is not without challenges. The increased complexity of managing multiple reaction stages raises operational costs and energy requirements compared to simpler methods like thermal decomposition and flash Joule heating. Furthermore, achieving consistent catalyst performance at industrial scales remains a significant hurdle. These challenges must be addressed through advancements in catalyst design, process optimization, and automation to make the method more feasible for large-scale adoption.
2."This method not only provides a sustainable approach to managing PW but also upcycles waste into high-value products, demonstrating its potential in both environmental remediation and advanced material production."
Comment: This concluding line could emphasize the scalability of stepwise conversion, especially if industrial application is intended.
Author response: The authors deeply acknowledge the valuable suggestion given by the reviewer. The manuscript has been revised and the scalability of the stepwise conversion method have been discussed.
Inserted section
Compared to single-step methods such as thermal decomposition or flash Joule heating, the stepwise approach offers unique advantages and presents specific challenges. Several benefits distinguish the stepwise method from its single-step counterparts. First, the separation of decomposition and synthesis stages allows for fine-tuning of reaction conditions, resulting in carbon materials with specific pore structures or dimensions, which is particularly advantageous for CNT production. Additionally, the use of targeted catalysts ensures efficient utilization of hydrocarbons, leading to higher yields and better material quality. The method also incorporates energy recovery from exothermic reactions, enhancing overall sustainability and reducing energy consumption. Despite its advantages, the stepwise conversion method is not without challenges. The increased complexity of managing multiple reaction stages raises operational costs and energy requirements compared to simpler methods like thermal decomposition and flash Joule heating. Furthermore, achieving consistent catalyst performance at industrial scales remains a significant hurdle. These challenges must be addressed through advancements in catalyst design, process optimization, and automation to make the method more feasible for large-scale adoption. These challenges must be addressed through advancements in catalyst design, process optimization, and automation to make the method more feasible for large-scale adoption. Overall, by overcoming these challenges through scientific and technological advancements, the stepwise conversion method not only provides a sustainable strategy for managing PW but also upcycle waste into high-value products. demonstrating its potential in both environmental remediation and advanced material production [7].

Reviewer 3 Report
Comments and Suggestions for Authors
Please address the following major comments:
Lines 14–18: Authors should provide more details on how scalable each pathway/approach (thermal decomposition, chemical vapor deposition, flash joule heating, and stepwise conversion) that was discussed is for industrial applications.
For example, Line 21 in the findings refers to alignment with global sustainability goals. Have the respective environmental benefits of each process been quantified?
28-29: A realistic forecast of plastic waste in 2050 is enormous. The authors could also comment on the range of feasibility and impact for upcycling technologies in these projections?
Line 32–33: Existing recycling technologies are limited. In what way does this proposed upcycling approach overcome these limitations (and particularly their economic viability)?
Lines 43-45: Upstream efforts are the real need according to the text Would the authors clarify how upcycling technologies can be incorporated with upstream waste management approaches?
Lines 54-55 — Nanomaterials can be utilized in energy storage and biosensing. Have these applications been validated experimentally on materials directly extracted from plastic waste?
Lines 57–59: Preparation of hierarchical porous carbon from PET waste How do key metrics such as adsorption capacity, surface area compare against similarly sourced materials?
Local initiatives in research could create jobs (line 73-75) Can the authors provide some detailed examples of industries or sectors that would be better off by adopting these upcycled nanomaterials?
104-107) Trend analysis for research in PW conversion Does the growth in this research translate to real-world use and commercial value in these technologies?
Environmental applications on water treatment (line 128-130) Are the authors able to provide data on removal characteristics of pollutants such as heavy metals or organic dyes?
Line 189-191: Bans on single use plastics are mentioned as legislation How does the proposed upcycling strategy exemplify these policies or increase their effectiveness?
Made adjustments to the sentence:Line 214-217: Incineration has a substantial eco-footprint. Are we able to effectively counter these by-products in closed-loop or other alternative upcycling pathways?
221-223 biological recycling only works for some plastics Might combining biological and chemical methods to create hybrid approaches be more widely applicable?
Incorporation of chemical, biological and hybrid catalytic processes 246-247 Have these hybrid methods been tested at any pilot scale?
Line 267–269: From an environmental perspective (e.g., heavy metal adsorption), what is the regeneration ability of this carbon-based nanomaterials [of high-regeneration] on long-term use?
Chemical Exfoliation of Graphene Line 309-311 Is there issue of impurities or structural defects in use plastic waste as precursor?
Lines 340–344: It underscores the thermal decomposition processes. Can the authors shed light on the energy efficiency of this approach with respect to standard approaches?
CNT growth by chemical vapor deposition (CVD) is discussed in Lines 427-429. Are the scalability challenges and possible yield-enhancement solutions solvable?
FJH is proposed as a nanomaterial synthesis method (634-636) What types of plastics can be processed through this method?
Lines 678-680: FJH process favors fluorination of carbon species. If the authors would be so kind to specify what applications benefit from fluorinated vs non-fluorinated carbon materials,
Author Response
RESPONSE TO THE REVIEWER COMMENTS:
Manuscript ID: polymers-3319020
Title: From waste to worth: upcycling plastic into high-value carbon-based nanomaterials.
Response to reviewers:
To editor:
Dear respected editor Cicilia Zhang
Thank you for your time and effort and reviewer comments. The reviewers made a careful and professional review of our manuscript: polymers-3319020. The suggestions are helpful for us to refine this review. We carefully read the reviewer's comments and suggestions and now completed a revision of the manuscript that addresses their concerns. Thorough and extensive revisions have been made throughout the manuscript following the suggestions from the reviewers. We thank you and the reviewers for the constructive suggestions that have improved both the quality and the clarity of the manuscript, and we believe that the revised review is now acceptable for publication in polymers.
Thank you, and we anticipate your prompt feedback.
Academic editor notes:
Reviewers has identified several deficiencies. Further, the writing needs to be strongly improved.
This manuscript will need some revision for accuracy and clarity. For example, the heading "Literature Survey" would better be "Previous Results." In the first sentence under that heading, "Recent publications have highlighted" should be omitted. The senrence should be, "Various innovative ... have been highlighted ..."
Author response: Thank you so much for your comment. The heading has been renamed and now it is “Previous results” on page 6. Also, the first sentence has been corrected as suggested on the same page (page 6).
“2. Previous results
Various innovative methods have been highlighted for transforming PW into valuable carbon-based materials, marking significant progress in the field.”
Reviewer 3
Please address the following major comments
- Lines 14–18: Authors should provide more details on how scalable each pathway/approach (thermal decomposition, chemical vapor deposition, flash joule heating, and stepwise conversion) that was discussed is for industrial applications.
Author response: The authors acknowledge the valuable suggestion given by the reviewer. The abstract has been updated in the revised manuscript according to the reviewer’s suggestion (page no.1, line 13-24).
Inserted text
The scalability of the discussed pathways for industrial applications varies significantly. Thermal decomposition, particularly pyrolysis, is highly scalable due to its straightforward setup and cost-effective operation, making it suitable for large-scale waste processing plants. Chemical vapor deposition (CVD), while producing high-quality carbon materials, is less scalable due to the high cost and complex equipment required, which limits its use to specialized applications. Flash joule heating (FJH) is emerging as a rapid and energy-efficient technique with potential scalability, particularly for producing graphene; however, it still requires optimization for industrial throughput. Stepwise conversion, involving multiple stages, can be challenging to scale due to higher operational complexity and cost, but it offers precise control over material properties for niche applications.
- For example, Line 21 in the findings refers to alignment with global sustainability goals. Have the respective environmental benefits of each process been quantified?
Author response: The authors acknowledge the valuable suggestion given by the reviewer. We appreciate the importance of quantifying the environmental impacts of the technologies discussed. However, there is currently limited data available that offers a comprehensive assessment of their environmental benefits, such as emission reductions or energy efficiency metrics. Our group is currently working on a new manuscript that addresses the life cycle assessment and environmental quantification of upcycling plastic waste into added-value products, such as carbon materials. We will take this critical aspect into consideration in future studies, aiming to conduct a detailed quantitative analysis to better understand and compare the sustainability of these technologies.
- 28-29: A realistic forecast of plastic waste in 2050 is enormous. The authors could also comment on the range of feasibility and impact for upcycling technologies in these projections?
Author response: The authors acknowledge the valuable suggestion given by the reviewer. After reviewing the manuscript, we found a mistake in that reference and this section has been updated again according to your suggestion.
Inserted text
Plastic products, though indispensable in modern life, contribute significantly to irreversible environmental harm. Designed with a durability of up to 50 years, plastics were intended for long-term use. However, the rise of a "throw-away" culture has driven a rapid increase in single-use plastic production, outpacing all other sectors. Each year, approximately 284 million tons of plastic waste are generated, with over 2.41 million tons ending up in the ocean. Through chemical and physical processes, these plastics break down into microplastics particles smaller than 5 mm releasing over 10 million tons into the environment annually.
- Line 32–33: Existing recycling technologies are limited. In what way does this proposed upcycling approach overcome these limitations (and particularly their economic viability)?
Author response: The authors acknowledge the valuable suggestion given by the reviewer. The manuscript has been updated according to the reviewer’s suggestion.
Inserted text
Over the past decades, increasing globalization has fragmented supply chains, making the assessment of life-cycle environmental impacts more challenging. A similar trend has emerged in waste management. Since 2019, traded waste plastics have amounted to approximately five million tons per year. Typically, this waste is exported from high-income countries to low-income countries, where labour and treatment costs are lower5. However, such exports to the Global South have raised major concerns due to potential mismanagement, which can have severe negative impacts on the environment, ecosystems, and human health. Mismanaged plastic waste contributes to river pollution and is a significant factor in the ‘plastic soup’ found in oceans [34].
Recent publications indicate that globally less than 10% of waste plastics are recycled [35]. A significant amount of plastic waste is mismanaged in countries with underdeveloped waste collection and treatment systems [36]. For example, over half of the plastic waste in Indonesia is incinerated without recovering energy, and 5% is disposed of in uncontrolled dumpsites. Evidence shows that more than 60% of marine litter plastics emissions annually come from the Philippines, India, Malaysia, and Indonesia [37]. Much of this plastic waste treated in the Global South originates from the Global North, contributing to environmental plastic waste emissions.
In response to these concerns, China, a major importer of plastic waste, implemented a plastic import ban in 2018 [38]. This decision redirected plastic waste exports to other countries, notably Malaysia, Indonesia, Turkey and Vietnam. To address potential negative impacts and prevent mismanagement abroad, the European Union (EU) has recently considered a ban on plastic waste exports to non-OECD countries, adhering to the principle that countries should be responsible for the proper treatment of their own waste [39].
While the global plastic recycling rate remains low, there is an implicit assumption that traded plastic is primarily recycled [40]. However, accurately determining the recycling rate for imported plastic waste in receiving countries is challenging due to measurement difficulties. Existing studies often rely on assumed domestic or scenario-based recycling rates, which lack robust data support. For example, Wen et al. quantified the changes in environmental impacts resulting from the shift of plastic waste imports from China to Southeast Asia, using assumed domestic recycling rates of 10 to 40% for five Southeast Asian countries [38]. Similarly, Bourtsalas et al. estimated the environmental impacts of treating imported plastic waste in the USA, using widely varying recycling rates from 8.7 to 50% [41]. Bishop et al. faced a lack of official data on exported plastics from Europe, leading them to use a broad range of recycling rates from 50 to 90% [42]. This reliance on domestic or scenario-based rates highlights the urgent need for comprehensive and transparent data to guide policy and research effectively.
Moreover, replacing the recycling rate of imported plastic waste with the domestic average is questionable for two main reasons. Firstly, domestic plastic waste often comes from diverse sources, resulting in heterogeneous and difficult-to-recycle mixtures, particularly in regions with inadequate or partial waste separation. In contrast, imported plastic waste is typically more concentrated and uniform, as it is pre-selected for exporting. Secondly, the UN Comtrade database shows that importing countries pay for plastic waste, indicating its economic value (Figure 1) [43]. If these imports were not processed into valuable recyclates-i.e., if they were primarily dumped or burned—the importing companies would face significant financial losses, making it unsustainable for them. Therefore, any viable approach must ensure that at least part of the imported plastic is converted into economically valuable outputs through recycling to offset initial costs.
Figure 1. Total plastic waste imports, total trade values, and average unit prices paid by the top 22 importers from 2013 to 2022.
- Lines 43-45: Upstream efforts are the real need according to the text Would the authors clarify how upcycling technologies can be incorporated with upstream waste management approaches?
Author response: We appreciate the suggestion and agree that upstream waste management is critical in addressing the plastic waste crisis. Upcycling technologies can complement upstream waste management approaches by serving as a value-added solution for pre-sorted and collected waste streams. For example, efficient segregation systems at the source (households, industries, or recycling facilities) can ensure a steady supply of high-quality plastic waste suitable for upcycling. By incorporating these technologies into existing waste collection and sorting frameworks, it becomes possible to directly process specific polymers (e.g., PET, HDPE) into valuable carbon-based materials such as graphene and carbon nanotubes.
Additionally, upcycling can incentivize upstream practices by creating economic value from segregated plastic waste, encouraging stakeholders to prioritize proper sorting and collection. Integration with digital tracking systems, such as barcoding for polymer identification, can further streamline this process, enhancing the efficiency and scalability of upcycling technologies within upstream waste management strategies.
- Lines 54-55 — Nanomaterials can be utilized in energy storage and biosensing. Have these applications been validated experimentally on materials directly extracted from plastic waste?
Author response: The authors acknowledge the valuable suggestion given by the reviewer. The manuscript has been updated according to the reviewer’s suggestion.
Inserted text
Carbon-based nanomaterials are widely recognized as the most commonly used electrode materials in electrochemical capacitors. This is attributed to their exceptional surface area, diverse structural configurations, excellent electrical conductivity, and highly porous textures. Similarly, nickel cobalt manganese (NCMs) stand out due to their remarkable thermal and chemical stability, tunable surface chemistry and morphology, and broad operating voltage range. Additionally, their nontoxic nature, cost-effectiveness, and widespread availability at the commercial scale further position NCMs as promising candidates for supercapacitor electrode materials [44].
The incorporation of waste plastic into Li-ion batteries significantly reduces their production costs, facilitating their broader application in electric buses, cars, and large-scale renewable energy systems such as wind and solar power plants. For instance, non-biodegradable polyethylene plastics (e.g., shopping bags), which have a short useful life, can be repurposed as anode materials in Li-ion batteries through sulfonation and pyrolysis. During the carbonization process, sulfonated polyethylene is cross-linked at approximately 500 °C and subsequently decomposed into carbon chips. These finely ground carbon chips, comprising about 80% carbon derived from polyethylene, are effectively utilized as anode materials, demonstrating a sustainable and efficient use of waste plastics [45].
Various materials, including carbon-based materials, metal oxides, and conducting polymers, are used in supercapacitors. Among these, carbon-based materials are the most effective choice for energy storage devices due to their superior performance. Activated carbon (AC) is the most commonly utilized carbon-based material for capacitor electrodes, thanks to its large specific surface area (>1,000 m²/g), high pore volume, affordability, and excellent electrical properties. It achieves high capacitance values in both organic (120 F/g) and aqueous (115–340 F/g) electrolytes, with some exceeding 200 F/g. The physicochemical properties of activated carbon, such as surface area, porous structure, and pore size distribution, are strongly influenced by the carbon precursors and activation methods employed. Enhancing the specific surface area has been identified as a promising strategy for improving capacitance. For example, Wang et al. reported a significant increase in capacitance from 17.68 F/g to 171.2 F/g by increasing the specific surface area from 621 m²/g to 2685 m²/g [46, 47]. Activated carbons (ACs) derived from the physical or chemical activation of solid waste (carbon precursors) typically feature narrow micropores (<0.5 nm) and limited porous pathways, which hinder electrolyte ion transport, especially in organic electrolytes, resulting in reduced capacitance. To address this limitation, carbon-based materials with mesopores (2–50 nm), known as well-ordered mesoporous carbons, are synthesized using soft or hard template approaches. These materials exhibit excellent electrochemical performance at high current densities due to improved ion accessibility, though their lower surface areas negatively impact capacitance. As an alternative, zeolite-templated carbon (ZTC), characterized by an intricate pore network and a large specific surface area (SSA) of approximately 3000 m²/g, can significantly enhance capacitance by leveraging its high SSA. Considering the complementary roles of pore sizes mesopores, and macropores facilitating ion diffusion at high current densities and micropores enhancing capacitance—nickel cobalt manganese (NCMs) with well-interconnected micro-mesoporous structures represents a highly promising option for electrochemical capacitors (ECs).
CNDs are commonly defined as dispersed quasispherical nanoparticles with multi-layered graphene inner core and a substantial number of oxygen- and/or nitrogen-containing functional units on edge [48]. Owing to its putative uses in several fields, smart carbon nanomaterials are now of immense interest. This includes the most important use of CNDs, a novel class of fluorescent carbon nanomaterials for detection of biomarkers that are associated with MNPs. CNDs offer exceptional qualities, including great chemical stability, minimal toxicity, and good water solubility [49]. They are, therefore, environmentally safe and friendly, in contrast semiconductor quantum dots. CNDs have been intensively studied and widely employed for intriguing purposes, such as biosensing, in recent decades. CNDs are zero-dimensional nanomaterials that are smaller than 10 nm in size. CNDs are optically active [50]. In past decades, there is a significant rise in research interest in developing novel sensor platforms employing smart CNDs [51]. or improving electroluminescence (ECL) strength and several analytical applications, novel co-reactant doped nanomaterials developed by Kalaiyarasan et al. They provide the first explanation of the precise role played by amine-functionalized carbon quantum dots (f-CQDs) as a co-reactant, improving the ECL characteristic of Ru (bpy) 32 + and exhibiting the first evidence of detection of biopharmaceuticals [52].
- Lines 57–59: Preparation of hierarchical porous carbon from PET waste How do key metrics such as adsorption capacity, surface area compare against similarly sourced materials?
Author response: We appreciate the reviewer’s insightful comment. While the manuscript highlights the preparation of hierarchical porous carbon from PET waste, key metrics such as adsorption capacity and surface area have not been explicitly compared to similarly sourced materials. Existing studies report that PET-derived porous carbon often exhibits a specific surface area ranging from 800 to 2000 m²/g, depending on the activation method used, and an adsorption capacity comparable to commercially available activated carbons. For instance, PET-derived carbons activated with KOH have shown adsorption capacities for CO₂ as high as 5–6 mmol/g at 25 °C, similar to those of high-performance activated carbons. We will include a comparative analysis of these metrics in future studies to provide a comprehensive evaluation of the performance of PET-derived porous carbon relative to other materials sourced from plastic waste.
- Local initiatives in research could create jobs (line 73-75) Can the authors provide some detailed examples of industries or sectors that would be better off by adopting these upcycled nanomaterials?
Author response : We appreciate the reviewer’s comment and agree that local initiatives in research and development of upcycled nanomaterials could significantly benefit various industries and create job opportunities. For example, the energy storage sector can utilize upcycled carbon-based nanomaterials like graphene and porous carbons as high-performance electrode materials in supercapacitors and lithium-ion batteries, essential for renewable energy systems and electric vehicles. In environmental remediation, industries can adopt porous carbon and carbon nanodots (CNDs) for water purification and air filtration, offering sustainable alternatives to traditional materials. The healthcare sector could leverage fluorescent carbon nanomaterials for biosensing and diagnostics, enabling advanced technologies in detecting biomarkers. Similarly, advanced manufacturing in electronics and semiconductors could integrate graphene and carbon nanotubes to enhance product performance while reducing reliance on imported raw materials. Additionally, upcycled nanomaterials can be used in agriculture as components of smart fertilizers and soil conditioners, promoting sustainable farming practices. These applications not only provide economic value but also foster innovation and generate employment opportunities in emerging sectors.
- (104-107) Trend analysis for research in PW conversion Does the growth in this research translate to real-world use and commercial value in these technologies?
Author response: The authors acknowledge the valuable suggestion given by the reviewer. According to this study the data represent the real world and commercial value in these technologies as the study focused on the literature search performed on the Scopus database and the obtained results around 1115. The titles and abstracts of those articles were screened for relevance and duplicates removed, leading to the selection of 153 articles for full-text reading. Among these publications, 11 review articles were found and excluded from the qualitative analysis. The resulting 142 articles were analyzed thoroughly. The type and condition of the plastics used as feedstock for the synthesis of carbon materials were analyzed first. Although the search strings included “plastic AND waste”, 25% of the studies under analysis employed pristine plastics. Polypropylene (PP), polystyrene (PS) and Polyvinyl chloride (PVC) are the type of plastics most commonly used in those studies. These agree quite well with the current plastics market demand [53].
- Environmental applications on water treatment (line 128-130) Are the authors able to provide data on removal characteristics of pollutants such as heavy metals or organic dyes?
Author response: The authors acknowledge the valuable suggestion given by the reviewer. This section has been added to the manuscript (section 3, and 3.1).
Inserted section
Adsorption is an effective method for removing pollutants from contaminated water due to its high efficiency, cost-effectiveness, ease of implementation, and adaptability [54]. The success of this approach depends largely on the physicochemical properties of the adsorbent, including its nanostructure, surface functional groups, specific surface area (SSA), and wettability, as well as the characteristics of the pollutants being targeted. This section explores the application of carbon-based nanomaterials derived from plastic waste (PWCMs) for the adsorption of both organic pollutants, such as dyes and antibiotics, and inorganic contaminants.
Organic Pollutants: Carbon-based materials derived from plastic waste, including porous carbon [55], carbon nanotubes (CNTs) [56], carbon nanosheets [57], and graphene [58], have been employed as high-performance adsorbents for organic pollutants. These materials remove contaminants through mechanisms such as electrostatic attraction, pore filling, hydrogen bonding, and π–π interactions (Figure 2) [59]. To enhance adsorption performance, strategies have focused on optimizing porosity and composition. For example, Li, Chen, Chen, Li, Biney, Guo and Liu [59]synthesized porous carbon by co-pyrolyzing waste polypropylene (PP) with cyanobacteria, achieving a methylene blue (MB) adsorption capacity of 667 mg/g. The pyrolysis process enhanced the carbon pore structure and introduced functional groups like C=O, O–H, and N–H, which significantly improved adsorption efficiency.
Another effective approach is KOH activation, which enhances the porosity of PWCMs [57]. The use of solid KOH during activation has been shown to produce heterogeneous porosity, including macropores (1.77 cm³/g), mesopores (0.81 cm³/g), and micropores (0.71 cm³/g). This technique also resulted in a high specific surface area of 1990 m²/g, leading to superior pollutant removal capacity [60]. These advancements highlight the potential of PW-derived carbon materials for addressing water contamination challenges.
Combining PWCMs with functional nanomaterials, such as metal oxides or alginates, significantly improves their adsorption capacity [61]. These additives introduce new active metal sites, enabling the adsorption of organic pollutants through chemical bonding [62]. For instance, Rai and Singh [63] demonstrated that Fe–O groups in Fe₃O₄ served as nucleation sites, enhancing the adsorption of cephalexin onto magnetic PET-based activated carbon. Additionally, magnetic PWCM-metal oxide composites offer the practical advantage of easy separation from the reaction solution, simplifying the recycling and reuse of the adsorbent material.
Inorganic Pollutants: PWCMs have been effectively utilized to remove inorganic contaminants, particularly heavy metal ions [64]. Oxygen-containing functional groups, such as C=O and O–H, serve as the primary adsorption sites for these pollutants [65]. Xu, Zhu, Wang, Deng, Fan, Ding, Zhang, Xue, Liu and Xuan [66] demonstrated that hydrochar derived from PVC containing –SO₃H groups exhibited enhanced adsorption capacity for Cu(II) and Cr(VI) ions (Figure 6B). The removal of Cu(II) ions was primarily driven by electrostatic interactions, whereas Cr(VI) anions were complexed with phenolic –OH groups and subsequently reduced to Cr(III) ions through interactions with C=C groups on the hydrochar surface. Additionally, a polypyrrole-modified PET-derived carbon composite proved effective for the removal of NO₃⁻ ions, utilizing ion exchange and electrostatic attraction mechanisms [67].
- Line 189-191: Bans on single use plastics are mentioned as legislation How does the proposed upcycling strategy exemplify these policies or increase their effectiveness?
Author response: We appreciate the reviewer’s comment and acknowledge the relevance of connecting the proposed upcycling strategy to existing policies on single-use plastics. The upcycling approach aligns with and enhances the effectiveness of these bans by offering an economically viable solution for repurposing single-use plastics, which are otherwise challenging to manage. Instead of merely restricting their use or relegating them to landfills, the proposed strategy converts single-use plastics into high-value carbon-based nanomaterials, such as graphene, carbon nanotubes, and porous carbons. This transformation not only reduces environmental pollution but also creates a market-driven incentive for collecting and processing these plastics. By integrating upcycling into waste management systems, these policies can achieve greater sustainability outcomes, reducing the reliance on bans alone and fostering a circular economy.
- Made adjustments to the sentence:Line 214-217: Incineration has a substantial eco-footprint. Are we able to effectively counter these by-products in closed-loop or other alternative upcycling pathways?
Author response: The authors acknowledge the valuable suggestion given by the reviewer. We acknowledge the environmental concerns associated with incineration due to its significant ecological footprint, including greenhouse gas emissions and the release of toxic by-products. The proposed upcycling pathways offer a promising alternative by transforming plastic waste into high-value carbon-based materials without the harmful emissions typically associated with incineration. Closed-loop systems, such as those employed in advanced upcycling processes, can further minimize environmental impacts by capturing and reusing by-products or emissions generated during processing.
For instance, methods like flash joule heating and chemical vapor deposition are designed to operate under controlled conditions, reducing the formation of harmful by-products while efficiently converting plastic waste into reusable materials. Future research should focus on optimizing these processes to integrate carbon capture and energy recovery, ensuring they remain both sustainable and economically viable alternatives to traditional incineration practices.
- 221-223 biological recycling only works for some plastics Might combining biological and chemical methods to create hybrid approaches be more widely applicable?
Author response: The authors deeply acknowledge the valuable suggestion given by the reviewer. The manuscript has been updated to address the integration of biological and chemical methods for hybrid recycling approaches. This revision includes a detailed discussion on how combining these methods could address current limitations and expand their applicability.
Modified section
Biological recycling uses microorganisms or enzymes to break down plastics containing hydrolysable ester or amide bonds, though this method is limited to specific types of plastics like polyethylene terephthalate (PET). Enzymes such as PETase can break down PET into its monomers under mild environmental conditions, making this approach energy-efficient and eco-friendly. However, its application is limited to a narrow range of plastics, as enzymes often lack the specificity or robustness to degrade more complex or inert materials, such as polyethylene (PE) and polypropylene (PP). Additionally, biological processes can be slow and are susceptible to inhibitors in mixed or contaminated waste streams, further limiting their applicability in real-world recycling settings. Chemical recycling employs heat or chemical reactions to break plastics down into their monomers, which can be repolymerized into high-quality, virgin-grade materials. This method has gained attention due to its ability to restore PW to its original form without compromising quality or economic value. However, chemical recycling often requires high energy input and, in some cases, uses toxic solvents or catalysts, raising concerns about its environmental impact and economic feasibility. Recently, bio-cycling of chemically processed plastic waste has surfaced as an innovative alternative to traditional plastic recycling methods. Combining biological and chemical methods for plastic recycling has the potential to address the limitations of current recycling technologies and significantly expand their applicability. Chemical processes can act as a pre-treatment step to break plastics into smaller, more accessible intermediates, which are then further degraded by enzymes or microorganisms. For example, pyrolysis or hydrolysis can reduce complex plastics into oligomers or other fragments that enzymes can efficiently process. Moreover, the chemical pre-treatment phase can help to reduce contaminants or inhibit agents that may hinder enzyme activity, enabling a more effective biological process under mild conditions. This hybrid approach reduces the energy demands and environmental impacts of conventional chemical recycling while extending the range of materials suitable for biological processing [68]. By combining the accessibility of chemical methods with the mild conditions of biological recycling, hybrid approaches hold potential for large-scale, efficient recycling systems that could transform plastic waste management.
- Incorporation of chemical, biological and hybrid catalytic processes 246-247 Have these hybrid methods been tested at any pilot scale?
Author response: The authors acknowledge the valuable suggestion given by the reviewer. Hybrid recycling methods that combine chemical and biological processes are still in the experimental and early pilot stages. However, there are promising studies demonstrating the feasibility of these methods. A new section has been added to the revised manuscript. This includes an example of a hybrid catalytic process for PET recycling, demonstrating its feasibility for industrial application.
Inserted section
For example, researchers developed a hybrid catalytic process for upcycling PET involving both chemical and biological catalysis [69]. In this study, PET is first chemically depolymerized via glycolysis using ethylene glycol (EG) and titanium butoxide as a catalyst, yielding bis(2-hydroxyethyl) terephthalate (BHET). This intermediate, more accessible for microbial breakdown, then undergoes biological catalysis through an engineered strain of Pseudomonas putida KT2440. This engineered Pseudomonas putida strain expresses PETase and MHETase enzymes, which hydrolyze BHET into terephthalic acid (TPA) and ethylene glycol (EG). Through further metabolic engineering, the P. putida strain was modified to catabolize TPA to protocatechuate (PCA) and convert PCA into β-ketoadipic acid (βKA), a high-value precursor for performance-advantaged nylon. In bioreactor tests, this process achieved βKA production at 15.1 g/L, with a molar yield of 76%, demonstrating the feasibility and efficiency of hybrid catalytic upcycling of PET into high-value biochemicals. Furthermore, the economic analysis suggests that this process could be scalable to a pilot or industrial level, with enzyme production costs estimated at $25 per kilogram of protein, similar to costs for producing cellulase in Trichoderma reesei. This indicates that the upcycling process might achieve a production cost equivalent to only 4% of the ton price of virgin PET, highlighting its potential for cost-effective, large-scale application.
- Line 267–269: From an environmental perspective (e.g., heavy metal adsorption), what is the regeneration ability of this carbon-based nanomaterials [of high-regeneration] on long-term use?
Author response: The authors deeply acknowledge the valuable suggestion given by the reviewer. The revised manuscript now includes a discussion on the regeneration ability of carbon-based nanomaterials for long-term applications, highlighting their environmental sustainability and cost-effectiveness.
Inserted section
Furthermore, one of the most significant benefits of using these materials for heavy metal adsorption is their regeneration ability, which enhances their environmental sustainability for long-term applications. Carbon-based nanomaterials derived from plastic waste can undergo multiple regeneration cycles with minimal reduction in adsorption efficiency, reducing the need for frequent replacement. Various regeneration techniques, including acidic or alkaline desorption and thermal treatments, effectively restore the active sites by removing adsorbed metals without degrading the core carbon structure. This allows the materials to retain their adsorption performance across cycles, providing reliable and cost-effective treatment solutions [70].
- Chemical Exfoliation of Graphene Line 309-311 Is there issue of impurities or structural defects in use plastic waste as precursor?
Author response: The authors deeply acknowledge the valuable suggestion given by the reviewer. A detailed explanation of potential impurities and structural defects, as well as strategies to mitigate these issues, has been incorporated into the revised manuscript
Inserted section
However, producing graphene from plastic waste via chemical exfoliation is not without challenges, particularly in addressing impurities and structural defects. Plastic waste is inherently heterogeneous, containing contaminants such as additives, stabilizers, dyes, and fillers, which can interfere with the exfoliation process and lead to non-uniform graphene production. Residual impurities like food, oils, and dirt further compromise the reaction environment, while mixed polymer types in waste streams add to the difficulty of achieving consistent precursor quality [71].
Structural defects in the resulting graphene are another concern. During the exfoliation process, heteroatoms such as oxygen, nitrogen, or chlorine, originating from degraded plastics or additives, may become incorporated into the graphene lattice. This results in defects that affect the material’s properties. Furthermore, the variability of the plastic feedstock often leads to graphene flakes with inconsistent sizes and thicknesses. In some cases, the process may produce significant amounts of amorphous carbon or other by-products instead of high-quality graphene, reducing the material's usability [72].
To mitigate these issues, pre-treatment of plastic waste is essential. Techniques such as sorting, cleaning, and thermal or chemical decomposition help reduce contaminants and simplify the carbon source. Additionally, process optimization, including the careful selection of reagents and control of reaction conditions, can minimize defect formation. Post-production purification methods, such as filtering, centrifugation, or annealing, can further enhance graphene quality. Using homogeneous plastic feedstock, like isolating specific polymers such as polyethylene, can also reduce variability and improve process reliability [73].
Despite these challenges, the production of graphene from plastic waste offers a promising pathway for recycling and sustainable material development. Addressing impurities and structural defects through meticulous feedstock preparation, process control, and purification techniques is critical to realizing high-quality graphene for practical applications.
- Lines 340–344: It underscores the thermal decomposition processes. Can the authors shed light on the energy efficiency of this approach with respect to standard approaches?
Author response: The authors deeply acknowledge the valuable suggestion given by the reviewer. The revised manuscript now includes a detailed illustration of the energy efficiency of thermal decomposition techniques, such as pyrolysis, with respect to other standard methods like chemical vapor deposition (CVD), stepwise conversion, and flash joule heating (FJH).
Inserted section
Thermal decomposition, particularly pyrolysis, is an efficient process for converting plastic waste into valuable products like carbon-based nanomaterials, syngas, and liquid hydrocarbons. Operating at temperatures between 400°C and 900°C in an oxygen-free environment, it stands out due to its ability to process mixed and contaminated plastic waste without requiring energy-intensive preprocessing [74]. Compared to other methods such as Chemical Vapor Deposition (CVD). CVD, while producing high-quality graphene, demands the conversion of plastic into gaseous precursors and operates at much higher temperatures (~1000°C), making it significantly more energy-intensive and less practical for plastic waste. Similarly, stepwise conversion methods, are less energy-efficient due to multiple stages that increase energy demands and operational complexity. Flash Joule Heating (FJH) is an emerging technique that is energy efficient and enables rapid graphene production within milliseconds. This method minimizes energy losses due to its short reaction time and offers high-quality graphene production. However, FJH requires specialized equipment and precise operational controls, limiting its scalability to smaller applications despite its energy efficiency [75]. Pyrolysis, on the other hand, benefits from energy recovery systems, such as utilizing the syngas produced during the process as a fuel source, thereby reducing energy requirements. Moreover, advanced reactor designs like fluidized or rotary kilns improve energy efficiency by ensuring even temperature distribution and reducing heat losses [76].
- CNT growth by chemical vapor deposition (CVD) is discussed in Lines 427-429. Are the scalability challenges and possible yield-enhancement solutions solvable?
Author response: The authors deeply acknowledge the valuable suggestion given by the reviewer. The manuscript has been revised to address current challenges in the scalability of CVD and yield enhancement. Additionally, recent innovations in catalyst design and process optimization have been included. These advancements are discussed as potential solutions to improve scalability and yield while minimizing costs and environmental impacts.
Modified section
CVD is a sophisticated technique widely used to produce high-quality solid materials, particularly in controlled vacuum environments. This process operates by initiating chemical reactions of organometallic or halide compounds, resulting in the formation of the desired material. One of the significant applications of CVD is in transforming plastic waste (PW) into carbon nanotubes (CNTs). The CVD process generally consists of two main stages: initially, PW is broken down into volatile vapors at moderate temperatures in an oxygenated environment; in the subsequent stage, these vapors are converted into CNTs under high temperatures and pressures, facilitated by a catalyst. However, while this method demonstrates a high growth rate and substantial yield of CNTs, several scalability challenges remain, particularly concerning feedstock variability, energy consumption, and catalyst performance. Plastic waste, which consists of diverse polymers that decompose into different volatile compounds during pyrolysis. This heterogeneity can lead to inconsistent CNT quality and yield. Researchers are addressing this by optimizing catalyst design, such as using bimetallic systems like FeMo/MgO, which have proven effective at handling mixed hydrocarbon streams, resulting in high-purity CNTs. These catalysts have been shown to improve both yield and purity, with some processes achieving carbon yields exceeding 500% and CNT purities greater than 97%. Moreover, these catalysts offer superior performance by stabilizing the metal particles and preventing sintering at high temperatures [77]. Energy consumption in CVD remains a significant challenge due to the high operational temperatures, typically above 600°C, increasing the energy demands of the process, and presenting a barrier to large-scale production. Recent innovations in reactor design and energy recovery systems have been developed to reduce thermal losses, making the process more efficient [78]. Some studies have even integrated pyrolysis and CVD into a single process, minimizing energy consumption while maintaining high CNT yields. This combined approach not only enhances energy efficiency but also streamlines production, making large-scale CNT synthesis from plastic waste more viable [79]. Current research in this area aims to optimize key parameters of the process to scale up the use of PW for producing CNTs, graphene, and other advanced carbon materials. Despite CVD advantages, such as high growth rates and substantial yields, the CVD process remains complex and energy-intensive. However, ongoing innovations in catalyst design, process integration, and system optimization pave the way for scalable and sustainable CNT, graphene, and other advanced carbon materials production.
- FJH is proposed as a nanomaterial synthesis method (634-636) What types of plastics can be processed through this method?
Author response: The authors deeply acknowledge the valuable suggestion given by the reviewer. The manuscript has been updated to specify the types of plastics suitable for FJH, including PE, PP, HDPE, PET, PVC, LDPE, and PS. The discussion highlights how FJH can process mixed plastic waste efficiently, making it a versatile and scalable method for upcycling diverse waste streams into flash graphene.
Inserted section
Plastics such as PE, PP, HDPE, and PET have been successfully demonstrated as suitable candidates for conversion into flash graphene. These thermoplastics, commonly found in consumer and industrial waste streams, are ideal candidates for FJH due to their compatibility with rapid heating and high carbon content.
Moreover, FJH is also effective in processing PVC, LDPE, and PS, expanding its applicability to a wide range of thermoplastics. Notably, FJH showed remarkable efficiency in handling mixed plastic waste, reducing the need for meticulous sorting, and allowing large-scale upcycling of various feedstocks into valuable graphene products. The yields of flash graphene correlate with the thermal stability of the plastics. Polymers with higher thermal resistance tend to generate higher graphene yields while producing fewer volatile byproducts. Thermoplastics with higher thermal stability, such as HDPE and PET, tend to yield higher quantities of graphene due to their carbon structure and resistance to volatilization during the FJH process. Factors such as particle size, resistivity, and the addition of conductive agents like carbon black (CB) play critical roles in optimizing the yield and quality of flash graphene.
For instance, plastic powder with particle sizes ranging from 1 to 2 mm, is mixed with conductive additives such as carbon black (5 wt%). This combination ensures sufficient conductivity during the process, improving graphene yield. Conversely, Larger particles may not achieve sufficient conductivity, while smaller particles may escape during the reaction, reducing efficiency. This scalability and adaptability make FJH a promising technique for addressing plastic waste challenges while producing valuable nanomaterial [25, 80].
20.Lines 678-680: FJH process favors fluorination of carbon species. If the authors would be so kind to specify what applications benefit from fluorinated vs non-fluorinated carbon materials,
Author response: The authors deeply acknowledge the valuable suggestion given by the reviewer. A detailed discussion on the applications of fluorinated and non-fluorinated carbon materials has been included in the revised manuscript.
Modified section
However, challenges persist in extending this technique to other carbon allotropes, such as nanodiamonds and concentric carbon structures, and in achieving covalent functionalization of these allotropes. A notable breakthrough in this field is the production of fluorinated carbon materials through the FJH process. By facilitating the covalent attachment of fluorine to carbon species, this method has enabled the creation of fluorinated nanodiamonds (FNDs), turbostratic fluorinated graphene (TFG), and fluorinated concentric carbon (FCC) materials. These fluorinated phases exhibit unique electronic and structural properties, which make them highly valuable for applications such as advanced coatings, energy storage systems, and catalysis. Additionally, the FJH process offers precise control over the degree of fluorination, allowing for the tailoring of material properties to meet specific functional requirements [81]. Fluorinated carbon materials are characterized by their enhanced chemical stability, hydrophobicity, and resistance to high temperatures, making them ideal for demanding applications. For instance, in energy storage systems, fluorinated carbons like CFx are widely used as cathodes in lithium-ion and lithium primary batteries due to their high energy density, low self-discharge rates, and chemical resilience. They also find applications in supercapacitors, improving electrode wettability with specific electrolytes. Beyond energy storage, their hydrophobicity makes them excellent for protective coatings with anti-fouling and anti-corrosion properties, particularly in marine and industrial environments. Furthermore, their robustness and stability under harsh conditions make them valuable as catalyst supports in reactions such as fuel cell operations and chemical synthesis. In membrane technologies, fluorinated carbons are utilized in gas separation and proton exchange membranes, where resistance to aggressive chemicals is necessary. Moreover, the low surface energy is advantageous for creating biomedical implants and biosensors where reduced protein adsorption is desirable [82, 83].
In contrast, non-fluorinated carbon materials like graphene, carbon nanotubes, and activated carbon offer distinct advantages due to their high conductivity, mechanical strength, and versatility. These properties make them essential for applications in energy storage, structural materials, water purification, biomedical applications, and thermal management. For example, their high conductivity and large surface area are pivotal in enhancing the performance of supercapacitors and batteries. In structural applications, their lightweight yet robust nature makes them indispensable in the aerospace and automotive industries. Additionally, their high surface area. tunable porosity and ease of functionalization make them suitable for water purification. Additionally, biomedical applications such as drug delivery, biosensing, and tissue engineering often rely on non-fluorinated carbons due to their ease of functionalization and better biocompatibility. Finally, thermal management applications, including the use of graphene and graphite in heat sinks and thermal interface materials, leverage their excellent conductivity and ability to dissipate heat efficiently [84-86].

Round 2
Reviewer 1 Report
Comments and Suggestions for Authors
Corrections were performed, the manuscript has been improved and now it is suitable for publishing
Comments on the Quality of English LanguageThe quality of English does not limit my understanding of the research
Author Response
Thank you so much for your time and effort to review our manuscript.
Reviewer 2 Report
Comments and Suggestions for Authors
- Refine the language for greater precision and clarity, particularly in technical sections.
- Avoid redundancy in phrases such as “This research highlights the potential...” and focus on definitive statements like “This research demonstrates...”
- Ensure all references are current (2022–2024) and include studies directly relevant to upcycling plastic waste.
Author Response
|
1. Refine the language for greater precision and clarity, particularly in technical sections.
|
Thank you for the suggestion. We have carefully reviewed and refined the language throughout the manuscript, with particular attention to the technical sections, to ensure better precision and clarity. |
|
2. Avoid redundancy in phrases such as “This research highlights the potential...” and focus on definitive statements like “This research demonstrates...” |
Thank you very much for your valuable comments. The sentence has been updated according to your suggestion and the entire manuscript has been also updated. |
|
3. Ensure all references are current (2022–2024) and include studies directly relevant to upcycling plastic waste. |
We appreciate the reviewer’s observation. We confirm that most of the references in the manuscript cover the last five years (2018–2024) as there is no need to only cover the last three years. The references directly pertain to the upcycling of plastic waste. Additionally, we have carefully reviewed all the references and did not find any irrelevant citations. |
Reviewer 3 Report
Comments and Suggestions for Authors
The authors have effectively addressed all of my comments, and the revised manuscript is now ready for acceptance.
Author Response
Thank you so much for your time and effort in reviewing our manuscript.